# SRSF3 promotes pluripotency through *Nanog* mRNA export and coordination of the pluripotency gene expression program

**Madara Ratnadiwakara[1,2,3†], Stuart K Archer[4], Craig I Dent[5], Igor Ruiz De Los Mozos[6], Traude H Beilharz[2,7], Anja S Knaupp[1,2,3], Christian M Nefzger[1,2,3], Jose M Polo[1,2,3], Minna-Liisa Anko[1,2,3†*]**

[1]Department of Anatomy and Developmental Biology, Monash University, Melbourne, Australia; [2]Biomedicine Discovery Institute, Development and Stem Cells Program, Monash University, Melbourne, Australia; [3]Australian Regenerative Medicine Institute, Monash University, Clayton, Australia; [4]Bioinformatics Platform, Monash University, Clayton, Australia; [5]School of Biological Sciences, Monash University, Melbourne, Australia; [6]The Francis Crick Institute, London, United Kingdom; [7]Department of Biochemistry and Molecular Biology, Monash University, Melbourne, Australia

**\*For correspondence:**
minni.anko@monash.edu

**Present address:** [†]Australian Regenerative Medicine Institute, Monash University, Clayton, Australia

**Competing interests:** The authors declare that no competing interests exist.

**Abstract** The establishment and maintenance of pluripotency depend on precise coordination of gene expression. We establish serine-arginine-rich splicing factor 3 (SRSF3) as an essential regulator of RNAs encoding key components of the mouse pluripotency circuitry, SRSF3 ablation resulting in the loss of pluripotency and its overexpression enhancing reprogramming. Strikingly, SRSF3 binds to the core pluripotency transcription factor *Nanog* mRNA to facilitate its nucleo-cytoplasmic export independent of splicing. In the absence of SRSF3 binding, *Nanog* mRNA is sequestered in the nucleus and protein levels are severely downregulated. Moreover, SRSF3 controls the alternative splicing of the export factor *Nxf1* and RNA regulators with established roles in pluripotency, and the steady-state levels of mRNAs encoding chromatin modifiers. Our investigation links molecular events to cellular functions by demonstrating how SRSF3 regulates the pluripotency genes and uncovers SRSF3-RNA interactions as a critical means to coordinate gene expression during reprogramming, stem cell self-renewal and early development.
DOI: https://doi.org/10.7554/eLife.37419.001

## Introduction

Induced pluripotent stem cells (iPSCs) can be generated from various cell types by the enforced expression of transcription factors such as OCT4, KLF4, SOX2 and MYC (OKSM) (*Takahashi and Yamanaka, 2006*). Reprogramming requires a complex coordination of cellular events to transition cells from a differentiated to a self-renewing fate. Many aspects of the molecular machinery governing pluripotency have been uncovered, with the focus on transcriptional regulation, and chromatin modifications. The master transcription factors OCT4, SOX2 and NANOG together with many other factors maintain pluripotency both in iPSCs and embryonic stem cells (ESCs) derived from the inner cell mass of the blastocyst stage embryo (*De Los Angeles et al., 2015*). Chromatin modifiers, such as the polycomb repressive complex 1 and 2 (PRC1/2), control self-renewal and differentiation potential by repressing lineage-specific genes and they reset the epigenetic landscape during reprogramming (*De Los Angeles et al., 2015*; *Takahashi and Yamanaka, 2006*).

RNA processing mediated by RNA-binding proteins (RBPs) encompasses many steps in the life of an RNA including pre-mRNA splicing, termination and polyadenylation, nonsense mediated decay (NMD) and mRNA export, with potential to significantly modify the gene expression output at each step. Global changes in alternative splicing (AS) and polyadenylation (APA) patterns take place during reprogramming (*Cieply et al., 2016*; *Ji and Tian, 2009*; *Lackford et al., 2014*; *Ohta et al., 2013*) but the functional significance of the differential RNA processing is not well understood. Remarkably, ESCs express more than 500 different RBPs (*Kwon et al., 2013*), the RBPs investigated to date highlighting the potential of RNA regulation in the control of pluripotency. For instance, MBNL1/2 splicing factors act as negative regulators of pluripotency and reprogramming (*Han et al., 2013*), the ectopic expression of the splicing factor ESRP1 enhances reprogramming (*Cieply et al., 2016*) and SRSF2 regulates alternative splicing in human pluripotent stem cells (*Lu et al., 2014*). FIP1 (encoded by *Fip1l1*) promotes maintenance of pluripotency by activating ESC-specific APA profiles (*Lackford et al., 2014*).

SRSF3 (serine-arginine-rich splicing factor 3) belongs to the family of SR protein splicing factors. We and others have previously shown that SRSF3 regulates constitutive and alternative splicing as well as other steps of RNA metabolism (*Änkö et al., 2012*; *Auyeung et al., 2013*; *Cavaloc et al., 1999*; *Huang and Steitz, 2001*; *Müller-McNicoll et al., 2016*; *Zhong et al., 2009*). At the cellular level, SRSF3 expression is regulated during cell cycle, SRSF3 overexpression in cancer cells enhances cell proliferation and its depletion leads to a cell cycle arrest (*Jia et al., 2010*; *Jumaa et al., 1997*; *Kurokawa et al., 2014*). SRSF3 plays a pivotal role during development as *Srsf3*-null mice fail to form blastocysts (*Jumaa et al., 1999*). SRSF3 was also identified as a potential regulator of reprogramming in an RNAi screen performed in mouse embryonic fibroblasts (MEFs) (*Ohta et al., 2013*), but the molecular or cellular mechanisms mediated by SRSF3 during reprogramming are not known. Despite the global identification of SRSF3 RNA targets (*Ankö et al., 2010*; *Änkö et al., 2012*; *Müller-McNicoll et al., 2016*), the functional significance of SRSF3 RNA-binding activity in vivo remains poorly characterised.

Here, we establish how SRSF3 governs pluripotency through its activities in RNA metabolism. Using individual-nucleotide resolution UV crosslinking and immunoprecipitation (iCLIP) and RNA sequencing (RNA-seq), we have identified a pluripotency circuitry directly regulated by SRSF3. We provide evidence for the critical role of SRSF3 in NANOG protein expression through its splicing-independent activity in the nucleo-cytoplasmic export of *Nanog* mRNA. However, SRSF3 function is not limited to regulating *Nanog*. To coordinate the pluripotency gene expression program, SRSF3 binds to multiple other (pre-)mRNAs encoding core pluripotency transcription factors, chromatin modifiers and RNA processing factors, including the export factor *Nxf1*. By combining SRSF3 RNA-binding sites with RNA-sequencing and functional data, we demonstrate that SRSF3 operates through multiple RNA processing mechanisms to tune and coordinate gene expression at the post-transcriptional level to establish and maintain pluripotency.

## Results

### SRSF3 facilitates reprogramming

To investigate the role of SRSF3 in reprogramming and self-renewal, we generated a reprogrammable tamoxifen inducible *Srsf3* knockout mouse model (*Srsf3*-KO/OKSM; *Figure 1A*). In cells isolated from these mice, reprogramming can be activated by Dox leading to reversible expression of OKSM factors, and Cre-recombinase can be induced by 4-hydroxy-tamoxifen (4OHT), resulting in the deletion of exons 2–3 of *Srsf3*. Cre-negative control MEFs reprogrammed with a high efficiency and gave rise to *bona fide* iPSCs capable of forming teratomas (*Figure 1—figure supplement 1A*), consistent with our previous report (*Alaei et al., 2016*). During reprogramming, *Srsf3* mRNA expression was upregulated first at day 3, followed by a sharp increase by day 9 (*Figure 1B*, dotted line). Analysis of several independent cell lines revealed significantly higher levels of *Srsf3* mRNA in ESCs and iPSCs compared to MEFs (*Figure 1—figure supplement 1B*). The biphasic increase in *Srsf3* expression coincided with the two transcriptional waves of reprogramming (*Polo et al., 2012*), where during the first wave the cell proliferation increases, lineage-specific genes are downregulated and major metabolic changes take place and during the second wave genes required for stem cell maintenance are activated. RNA-sequencing data showed an increase in *Srsf3* mRNA expression specifically in

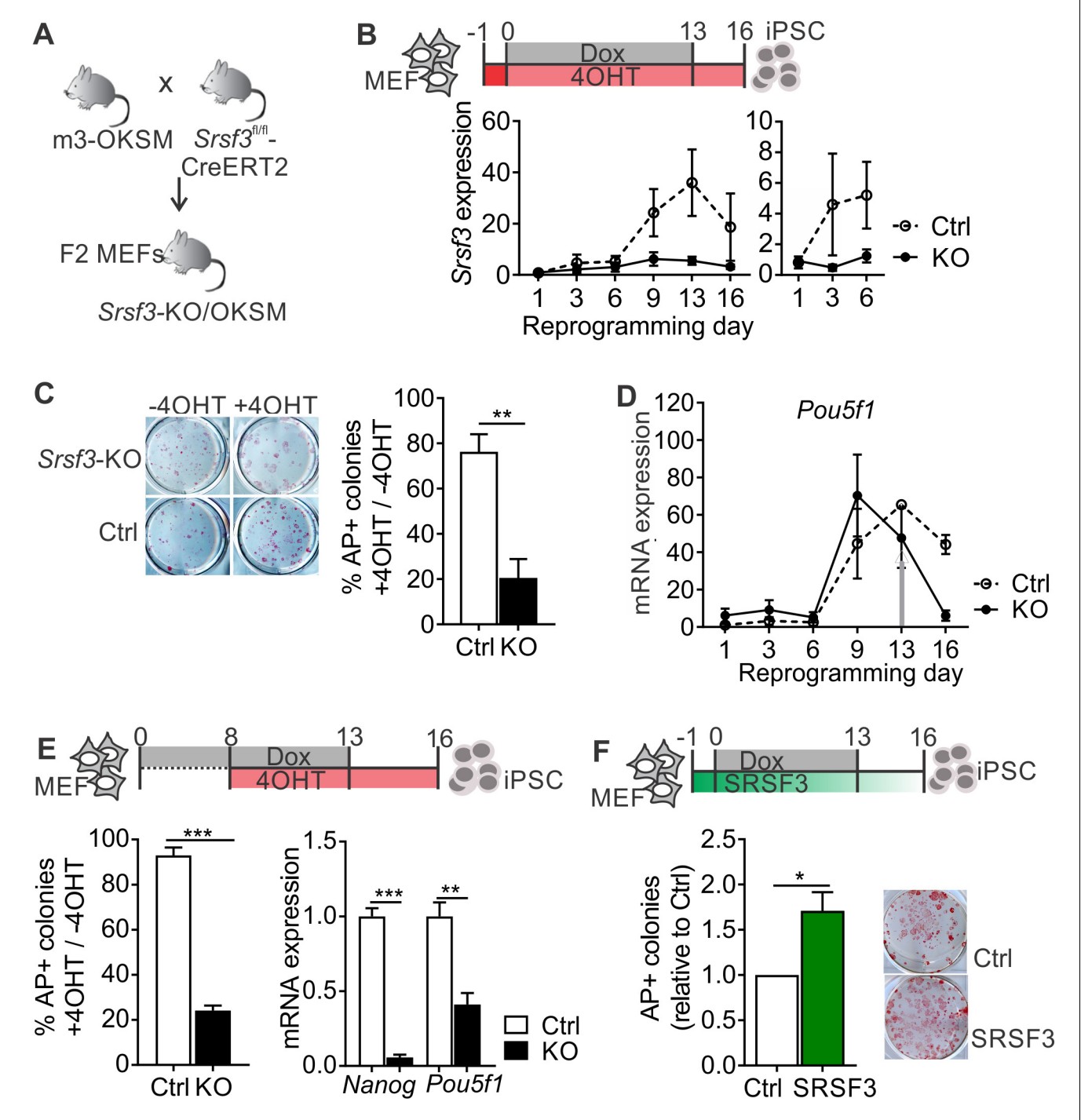

**Figure 1.** SRSF3 is essential for reprogramming. (A) The breeding strategy to obtain reprogrammable mice with a conditional *Srsf3* knockout allele (*Srsf3*-KO/OKSM). (B) Experimental outline of the reprogramming experiments (*top*). Quantification of *Srsf3* mRNA levels by RT-qPCR in SRSF3 depleted (KO) and control (Ctrl) cells in the course of reprogramming from day 1 to day 16 (*left*). Days 1–6 are shown on the right to visualise the expression changes within the first phase of reprogramming. Two-way ANOVA (***p=0.0002 for genotype, ***p=0.0008 for time and *p=0.0154 the interaction, data as mean ± SEM, n = 3–5). (C) Alkaline Phosphatase (AP) labelling of SRSF3 deficient (KO) and control (Ctrl) cells at day 16 of reprogramming (*left*) and quantification of AP-positive colonies (*right*). The data is presented as a relative number of colonies between treated (+4 OHT) and untreated (−4OHT) cells reprogrammed in parallel (Unpaired Student's t-test, two-tailed, **p<0.01, data as mean ± SEM, n = 4–5). (D) Quantification of *Pou5f1* mRNA expression by RT-qPCR during reprogramming in SRSF3 depleted (KO) and control (Ctrl) cells. The grey arrow denotes the point of Dox withdrawal and start of endogenous *Pou5f1* expression (data as mean ± SEM, n = 2). The data is normalised to *Hprt* and presented relative to control MEFs. (E) Experimental outline (*top*) of experiments where MEFs were reprogrammed for 8 days before SRSF3 depletion.

*Figure 1 continued on next page*

*Figure 1 continued*

Quantification of AP positive colonies (*left*) and pluripotency marker expression (*right*) at day 16 of reprogramming as in (C). The AP counts and mRNA expression are presented as above (Unpaired Student's t-test, two-tailed, \*\*p<0.01, data as mean ± SEM, n = 4). (F) Experimental outline of the SRSF3 overexpression experiments (*top*). AP labelling of colonies at day 16 of reprogramming and quantification of AP-positive colonies. The data is presented relative to GFP-only (Ctrl) transfected cells (Unpaired Student's t-test, two-tailed, \*p<0.05, data as mean ± SEM, n = 3). See also *Figure 1— figure supplement 1*.

DOI: https://doi.org/10.7554/eLife.37419.002

The following figure supplement is available for figure 1:

**Figure supplement 1.** SRSF3 depletion and overexpression have the opposite effects on reprogramming outcome.

DOI: https://doi.org/10.7554/eLife.37419.003

cells that successfully formed iPSCs compared to cells refractory to reprogramming (*Polo et al., 2012*) (*Figure 1—figure supplement 1C*).

To determine how SRSF3 depletion affects reprogramming efficiency, *Srsf3*-KO/OKSM MEFs were first treated with 4OHT for 24 hr before inducing the OKSM transgene expression with Dox. The 4OHT exposure resulted in a sustained reduction of *Srsf3* mRNA (*Figure 1B*, solid line), with no effect on control cells. After removal of Dox at day 13, the cells were cultured for an additional 3 days to generate transgene independent iPSCs. The iPSC colonies were detected by alkaline phosphatase (AP) labelling both at day 13 (prior to Dox removal) and at day 16. The numbers of AP positive colonies were significantly reduced in SRSF3-depleted cells compared to controls and untreated (−4OHT) cells (*Figure 1C* and *Figure 1—figure supplement 1D*). The few AP-positive colonies that were generated from SRSF3-deficient MEFs consistently failed to form teratomas (n = 5, *Figure 1— figure supplement 1E*). The failure of SRSF3-depleted cells to induce the pluripotency program was further confirmed by a strongly reduced expression of *Nanog* and endogenous *Pou5f1* (encoding OCT4) from day 13 onwards when Dox - and ectopic OKSM expression - was removed (*Figure 1D* and *Figure 1—figure supplement 1F*). We did not detect significant increase in cell death in SRSF3-deficient MEFs during reprogramming (*Figure 1—figure supplement 1G*), similar to previous observations (*Ohta et al., 2013*). As the first phase of reprogramming involves the activation of cell proliferation (*Polo et al., 2012*), we also assessed cell proliferation in SRSF3-deficient reprogramming cells. We detected a systematic reduction in the number of SRSF3-depleted cells when compared to controls, whereas SRSF3 overexpression led to an increase in cell numbers (*Figure 1—figure supplement 1H*).

To assess the role of SRSF3 in the activation of the pluripotency program during the second phase of reprogramming, we first reprogrammed the cells for 8 days after which SRSF3 depletion was induced and AP-positive colonies detected at day 16. SRSF3 was efficiently depleted following 4OHT treatment by day 9 (*Figure 1—figure supplement 1I*) and the number of transgene-independent AP positive colonies was significantly reduced at day 16, accompanied by a reduced expression of pluripotency markers (*Figure 1E*). These data demonstrate that SRSF3 is required to activate the pluripotency program during reprogramming.

Finally, we determined if SRSF3 overexpression could improve the reprogramming outcome. Reprogrammable MEFs were transduced with a vector encoding *Srsf3* and a GFP reporter or only GFP (*Figure 1F* and *Figure 1—figure supplement 1J*). Twenty-four hours later, reprogramming was induced with Dox and transgene-independent iPSC colonies were detected by AP labelling at day 16. SRSF3 overexpression in MEFs resulted in a significantly greater number of AP-positive colonies compared to GFP only controls (*Figure 1F*). Flow cytometric analysis of cell surface marker expression at reprogramming days 6 and 16 further supported the enhanced reprogramming efficiency (*Figure 1—figure supplement 1K*). This analysis demonstrates that SRSF3 is not only essential for reprogramming but is also able to boost the generation of AP-positive colonies.

## SRSF3 is essential for the maintenance pluripotency

To further address the role of SRSF3 is pluripotent cells, we generated iPSCs from *Srsf3*-KO/OKSM MEFs without 4OHT exposure and cultured them for multiple passages before Cre activation (*Figure 2A*). Light microscopic inspection showed a rapid loss of iPSC colony morphology following SRSF3 depletion (*Figure 2B*). NANOG protein expression was strongly reduced in SRSF3-deficient iPSCs (*Figure 2C*), supporting the exit from pluripotency. Surprisingly, *Nanog* mRNA levels were

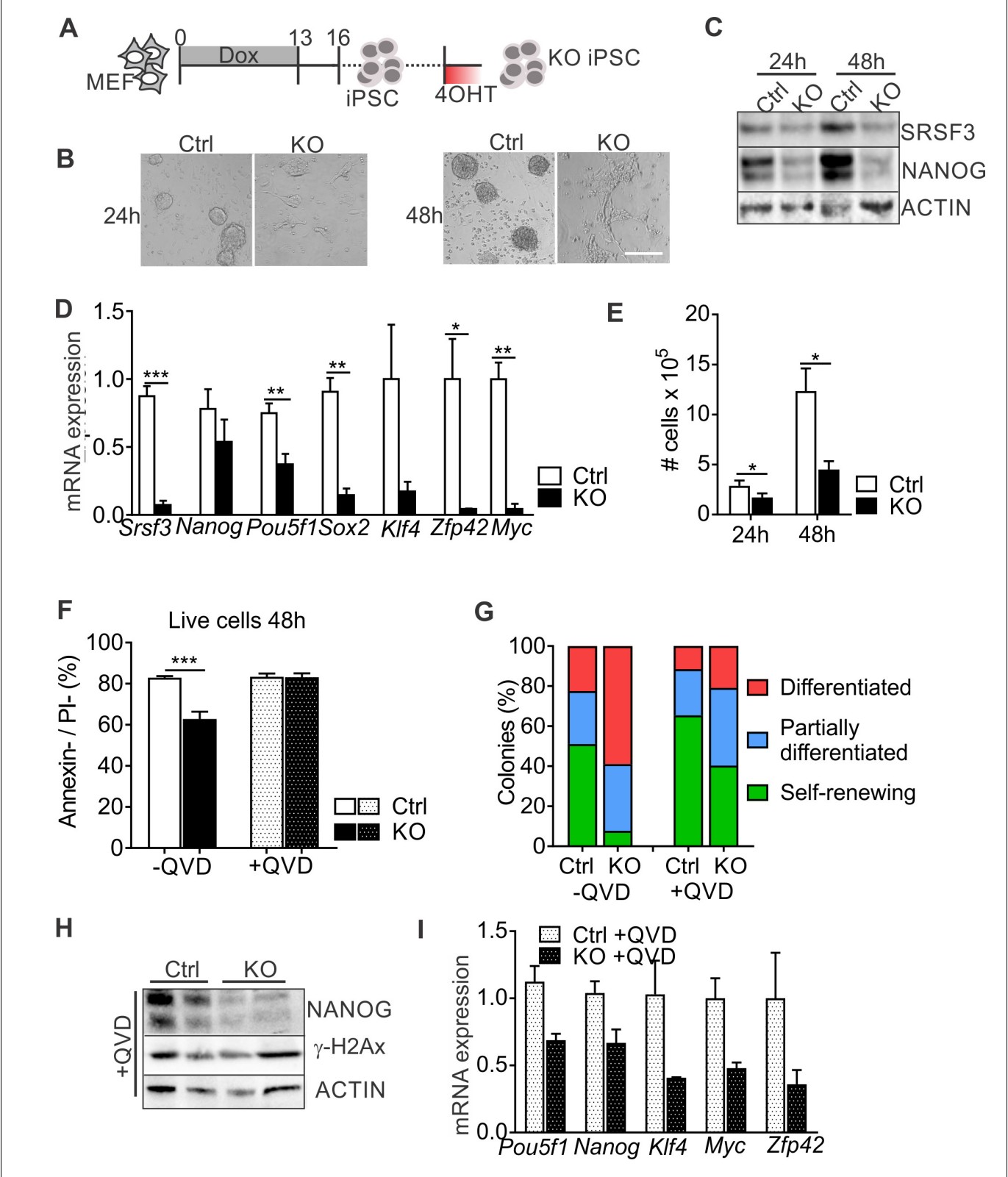

**Figure 2.** SRSF3 depletion leads to loss of pluripotency independent of cell death. (**A**) Experimental outline of the *Srsf3*-KO iPSC generation. (**B**) Phase contrast images of SRSF3 depleted (KO) and control (Ctrl) iPSCs 24 and 48 hr after 4OHT induction. Scale bar: 50 µm. (**C**) Western blot analysis of SRSF3 and NANOG expression in SRSF3 deficient (KO) and control (Ctrl) iPSCs 24 and 48 hr after 4OHT induction. ACTIN served as a loading control. (**D**) Quantification of *Srsf3* and pluripotency marker mRNA expression in *Srsf3*-KO (KO) and control (Ctrl) iPSCs 24 hr after 4OHT induction. The data is

*Figure 2 continued on next page*

*Figure 2 continued*

normalised to *Hprt* and presented relative to control iPSCs (Unpaired Student's t-test, two-tailed, *p<0.05, **p<0.01, ***p<0.001, data as mean ± SEM, n = 3). (E) Quantification of cell numbers 24 and 48 hr after 4OHT induction in *Srsf3*-KO (KO) and control (Ctrl) iPSCs (Unpaired Student's t-test, two-tailed, *p<0.05, data as mean ± SEM, n = 4). (F) Quantification of live cells in untreated and QVD treated *Srsf3*-KO (KO) and control (Ctrl) iPSCs by AnnexinV/PI labelling and flow cytometry 48 hr after 4OHT induction (Unpaired Student's t-test, two-tailed, *p<0.05, data as mean ± SEM, n = 3). (G) Quantification of pluripotent and differentiated iPSC colonies in untreated and QVD treated *Srsf3*-KO (KO) and control (Ctrl) iPSCs. Colonies were scored from three independent experiments. (H) Western blot analysis of γH2Ax and NANOG expression in QVD treated *Srsf3*-KO (KO) and control (Ctrl) iPSCs 48 hr after 4OHT induction. ACTIN served as a loading control. (I) Quantification of *Srsf3* and pluripotency marker mRNA expression in QVD treated *Srsf3*-KO (KO) and control (Ctrl) iPSCs 24 hr after 4OHT induction. The data is normalised to *Hprt* and presented relative to control iPSCs (data as mean ± SEM, n = 2). See also *Figure 2—figure supplement 1*.

DOI: https://doi.org/10.7554/eLife.37419.004

The following figure supplement is available for figure 2:

**Figure supplement 1.** Apoptosis inhibition in SRSF3 depleted iPSCs.

DOI: https://doi.org/10.7554/eLife.37419.005

little affected while the expression of other pluripotency genes such as *Pou5f1* (encoding OCT4), *Sox2*, *Klf4*, *Zfp42* (encoding REX1) and *Myc* was strongly reduced (*Figure 2D*). Similar to MEFs, SRSF3 depletion in iPSCs led to a reduction in cell numbers (*Figure 2E*). Whereas SRSF3-deficient MEFs showed no cell death, the level of AnnexinV/PI-negative cells was reduced in SRSF3-depleted iPSCs compared to controls, indicative of an increase in apoptosis (*Figure 2F*). To exclude that the loss of pluripotency in SRSF3-deficient cells was due to apoptosis, we treated SRSF3 depleted and control iPSCs with the caspase inhibitor QVD and assessed pluripotency (*Figure 2F*). Even in the absence of apoptosis, SRSF3 depletion led to a stark increase in the number of differentiating colonies and reduction in the expression of pluripotency markers (*Figure 2G–I*, *Figure 2—figure supplement 1A–B*), demonstrating that SRSF3 directly contributed to the maintenance of pluripotency. In support of this and unlike SRSF1 and SRSF2 (*Li and Manley, 2005*; *Xiao et al., 2007*), SRSF3 depletion did not result in increased γH2Ax levels marking DNA double stranded breaks (*Figure 2H*), suggesting that SRSF3 loss does not lead to genomic instability. These data demonstrate an essential role for SRSF3 in self-renewing cells, loss of SRSF3 leading to exit from pluripotency independent of cell death.

## iCLIP reveals SRSF3 binding to RNAs encoding key factors of the pluripotency circuitry

To determine if SRSF3 directly regulates the expression of pluripotency factors, we performed individual-nucleotide-resolution UV crosslinking and immunoprecipitation (iCLIP) (*Figure 3A*) (*Änkö et al., 2012*; *Huppertz et al., 2014*). First, we generated ESCs expressing GFP-tagged SRSF3 at an endogenous level from a BAC transgene (*Ankö et al., 2010*). Nuclear GFP expressing ESCs served as controls (*Figure 3—figure supplement 1A–B*, *Figure 3—source data 1–2*). In parallel, we also performed iCLIP using an antibody against the endogenous SRSF3. The pooled SRSF3-GFP iCLIP reads yielded 3.3M unique crosslink sites compared to 51K in the control while the anti-SRSR3 iCLIP resulted in low read coverage, suggesting a poor affinity of the antibody to RNA-bound SRSF3 (*Figure 3—source data 3*, *Figure 3—source data 4*). Significant crosslink sites were identified as previously described (FDR < 0.05, (*Änkö et al., 2012*; *König et al., 2010*)), resulting in 219K SRSF3-GFP sites in the pluripotent transcriptome (*Figure 3—source data 4*). The consensus binding motif was consistent with previous SRSF3-GFP iCLIP and in vitro studies (*Figure 3B*) (*Änkö et al., 2012*; *Hargous et al., 2006*; *Müller-McNicoll et al., 2016*). The consensus binding motifs of anti-SRSF3 and SRSF3-GFP iCLIPs were identical (*Figure 3—figure supplement 1C*) and there was a high overlap (74%) between anti-SRSF3 and SRSF3-GFP CLIP sites, consistent with previous studies demonstrating the functionality of SRSF3-GFP (*Ankö et al., 2010*). Because SRSF3-GFP resulted in ~100 fold higher number of unique cDNAs than the anti-SRSF3 iCLIP, we only proceeded with the analysis of SRSF3-GFP iCLIP data. SRSF3-binding sites were distributed over various RNA segments, including exons and introns both in coding and non-coding RNAs (*Figure 3C*). Gene Ontology (GO) analysis of genes with significant SRSF3 binding sites showed an enrichment in functions related to RNA metabolism (*Figure 3—figure supplement 1D*), supporting previous observations (*Änkö et al., 2012*; *Müller-McNicoll et al., 2016*). Importantly, we detected an enrichment across GO terms

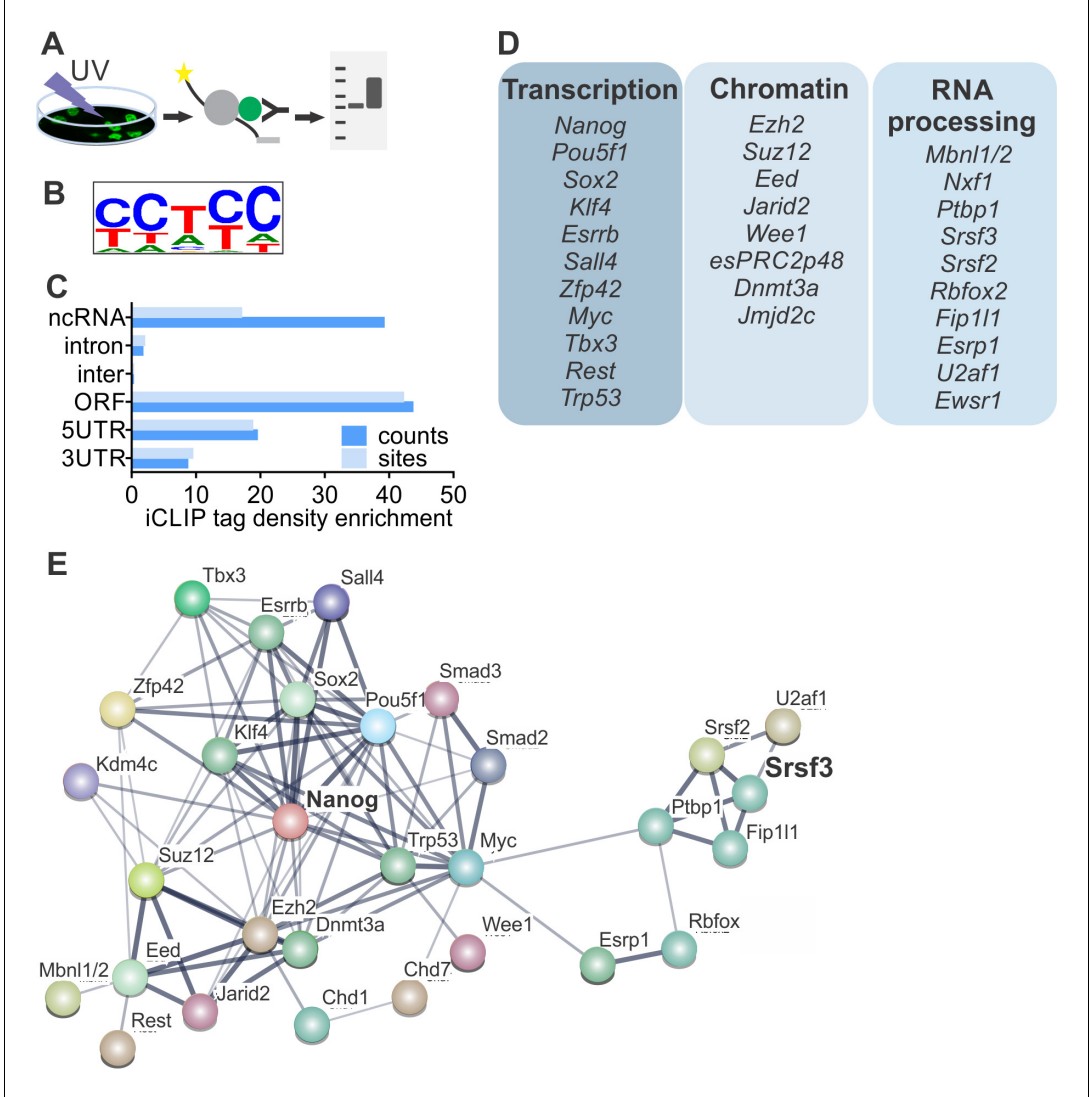

**Figure 3.** SRSF3 binds to RNAs encoding key pluripotency factors. (**A**) Schematic of iCLIP to determine SRSF3 binding sites in pluripotent cells. (**B**) SRSF3 consensus binding motif in pluripotent cells. (**C**) Distribution of significant SRSF3 crosslink sites (FDR < 0.05) over different transcript regions normalised to feature length. (**D**) SRSF3 binding sites detected in RNAs encoding key pluripotency regulators of multiple functional categories. (**E**) Functional association network of SRSF3 and its RNA targets in pluripotent cells. No connection between SRSF3 and the NANOG centred pluripotency network has been previously assigned. Pluripotency regulators that were directly bound by SRSF3 were used as input for STRING (*Szklarczyk et al., 2015*). The pluripotency associated transcription factors and chromatin modifiers were based on *Young (2011)* and mRNA processing factors based on *Chen and Hu (2017)*. The thickness of the line reflects the confidence level for the interaction (highest 0.9 – high 0.7 – medium 0.4 – low 0.15), large nodes represent proteins with some structural information. See also *Figure 3—figure supplement 1* and *Figure 3—source data 1–4*.

DOI: https://doi.org/10.7554/eLife.37419.006

The following source data and figure supplement are available for figure 3:

**Source data 1.** iCLIP peaks for SRSF3-EGFP.
DOI: https://doi.org/10.7554/eLife.37419.008
**Source data 2.** iCLIP peaks for EGFP-NLS.
DOI: https://doi.org/10.7554/eLife.37419.009
**Source data 3.** iCLIP peaks for anti-SRSF3.
DOI: https://doi.org/10.7554/eLife.37419.010
**Source data 4.** SRSF3 iCLIP mapping statistics.
DOI: https://doi.org/10.7554/eLife.37419.011
**Figure supplement 1.** SRSF3 iCLIP in pluripotent cells.
DOI: https://doi.org/10.7554/eLife.37419.007

associated with chromatin modifications, transcriptional regulation, cell division/cycle and DNA damage response (*Figure 3—figure supplement 1D*), which are central functions in the establishment and maintenance of pluripotent cells.

Guided by the GO analysis, we next analysed SRSF3-binding sites within the pluripotency circuitry. SRSF3 bound to a large number of (pre-)mRNAs encoding pluripotency transcription factors (*Figure 3D*), including *Nanog, Pou5f1, Zfp42, Klf4, Sox2* and *Myc* that were downregulated in SRSF3 depleted iPSCs (see *Figure 2D*). Significant SRSF3-binding sites were also detected in RNAs encoding chromatin modifiers (*Figure 3D*) such as the components of PRC2 required to deposit repressive histone-3 lysine-27 tri-methyl (H3K27me3) marks in self-renewing cells. Interestingly, SRSF3 bound to RNAs encoding RBPs with previously assigned roles in reprogramming and/or pluripotent cells (*Figure 3D*). For example, MBNL1/2 splicing factors have been shown to act as negative regulators of pluripotency and reprogramming (*Han et al., 2013*) and FIP1 (encoded by *Fip1l1*) to promote the maintenance of pluripotency (*Lackford et al., 2014*). Functional protein association network analysis by STRING (*Szklarczyk et al., 2015*) showed that the pluripotency related - as defined by (*Chen and Hu, 2017*; *Young, 2011*) - SRSF3 RNA targets formed an interconnected network while SRSF3 has not been previously assigned as part of this network (*Figure 3E*), demonstrating that our iCLIP analysis determined hitherto uncharacterised regulatory interactions in pluripotent cells.

## SRSF3 regulates NANOG protein levels by mediating the export of *Nanog* mRNA

The genome wide-analysis suggested that SRSF3 directly controls a NANOG-centred pluripotency network. To determine the molecular mechanism underlying SRSF3 function in pluripotency, we first turned our attention to NANOG, the core transcriptional regulator in pluripotent stem cells. We wondered why NANOG protein expression was drastically reduced, whereas *Nanog* mRNA expression was largely unaffected in SRSF3 deficient iPSCs (*Figure 2C–D*). Although *Nanog* gene produces three annotated splice variants, only one of the variants was significantly expressed in iPSCs based on our RNA-seq data (*Figure 4A*). This variant encodes the full-length (FL) NANOG protein, whereas the two other splice variants produce an N-terminally truncated (TR) isoform. RT-qPCR analysis showed that SRSF3 depletion did not affect the first exon choice of *Nanog* mRNA (*Figure 4B*). A previous study in P19 cells predicted *Nanog* as one of ~400 mRNAs whose nucleo-cytoplasmic export was affected by SRSF3 knockdown (*Müller-McNicoll et al., 2016*). The direct interaction of SRSF3 with the nuclear export factor NXF1 is well-documented (*Huang et al., 2003*; *Huang and Steitz, 2001*; *Müller-McNicoll et al., 2016*). Accordingly, subcellular fractionation revealed nuclear accumulation of *Nanog* mRNA in SRSF3 depleted iPSCs (*Figure 4C* and *Figure 4—figure supplement 1A–C*).

The subcellular fractionation analysis together with the robust reduction in NANOG protein levels following SRSF3 depletion in iPSCs (see *Figure 2C*) suggested that SRSF3 regulates NANOG protein levels through its mRNA export adaptor activity. To investigate mechanistically if direct SRSF3 binding to *Nanog* mRNA was required for its export, we generated *Nanog* expression constructs carrying either the wild type *Nanog* cDNA sequence including 3'UTR (hereafter WT-*Nanog*) or a *Nanog* cDNA sequence where silent mutations abolishing SRSF3-binding sites had been introduced (ΔSRSF3-*Nanog*) (*Figure 4D*). Following transfection into pluripotent cells, these constructs were expressed at similar levels, which did not exceed the endogenous levels (*Figure 4—figure supplement 1D*). As expected, RNA immunoprecipitation showed strongly reduced binding of SRSF3 to ΔSRSF3-*Nanog* mRNA (*Figure 4E*). Nucleo-cytoplasmic fractionation demonstrated that ΔSRSF3-*Nanog* mRNA lacking SRSF3-binding sites was retained in the nucleus both in control and SRSF3-deficient cells (*Figure 4F*, dotted bars). The intronless mRNA produced from the WT-*Nanog* construct was efficiently exported in control cells while it was retained in the nucleus in SRSF3-depleted iPSCs (*Figure 4F*, compare white and black bars). This demonstrates firstly, that SRSF3 binding is essential for *Nanog* mRNA export and secondly, that SRSF3 activity in *Nanog* mRNA export is independent of pre-mRNA splicing. We also transfected the WT-*Nanog* or ΔSRSF3-*Nanog* expression construct into HEK293 cells that do not express endogenous *Nanog* and do not functionally depend on NANOG protein. Similar to pluripotent cells, SRSF3 bound to WT-*Nanog* but not ΔSRSF3-*Nanog* mRNA (*Figure 4G*, right αGFP IP). RNA immunoprecipitation with an anti-NXF1 antibody showed NXF1 binding to WT-*Nanog* mRNA while in the absence SRSF3-binding sites the *Nanog*-NXF1 interaction was abolished (*Figure 4G*, left αNXF1 IP; *Figure 4—figure supplement 1E*), demonstrating

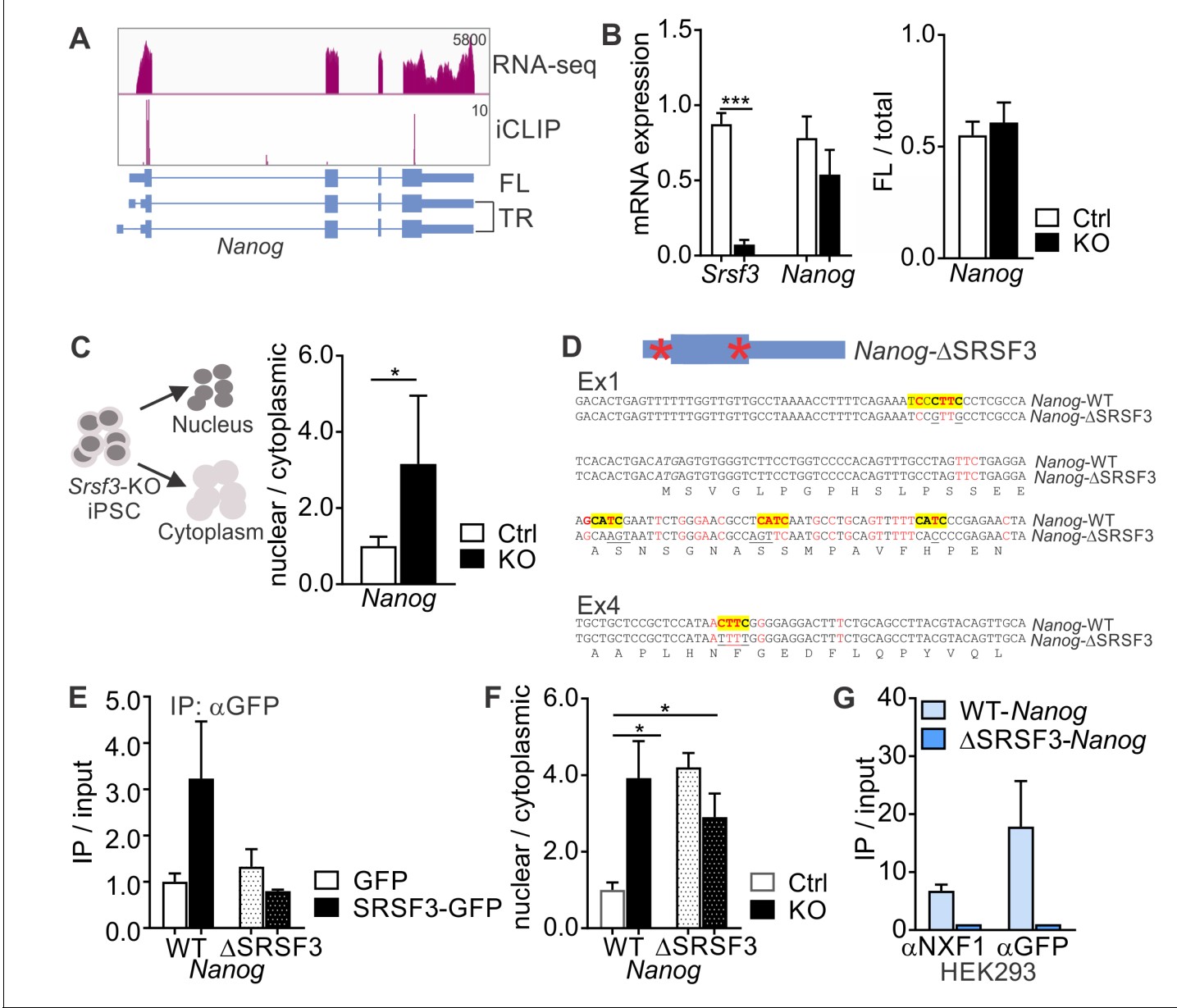

**Figure 4.** SRSF3 binding and mRNA export adaptor activity in independent of splicing and essential for *Nanog* mRNA export. (**A**) *Nanog* expression and SRSF3 iCLIP binding profile within *Nanog* transcript region. Gene diagram depicting *Nanog* variants is presented below the graphs (FL = full length, TR = truncated). (**B**) *Left:* RT-qPCR quantification of *Srsf3* and *Nanog* mRNA expression in *Srsf3*-KO (KO) and control (Ctrl) iPSCs 24 hr after 4OHT induction. The data is normalised to *Hprt* and presented relative to control iPSCs (***p<0.001, Unpaired Student's t-test, two-tailed, data as mean ± SEM, n = 3). *Right:* Quantification of FL *Nanog* transcript levels relative to total *Nanog* mRNA levels (p>0.05, Unpaired Student's t-test, two-tailed, data as mean ± SEM, n = 3). (**C**) *Srsf3*-KO (KO) and control (Ctrl) iPSCs were fractionated into nuclear, cytoplasmic and total fractions 24 hr after 4OHT induction (*left*). Quantification of *Nanog* mRNA levels in the nuclear and cytoplasmic fractions (*right*). Data is presented as nuclear to cytoplasmic ratio (*p<0.05, Unpaired Student's t-test, two-tailed, data as mean ± SEM, n = 5) and normalised to control. #error bar smaller than the border line. (**D**) Generation of WT-*Nanog* and ΔSRSF3-*Nanog* expression constructs. Both constructs contain the complete *Nanog* cDNA, including 5' and 3' UTRs. In ΔSRSF3-*Nanog*, synonymous mutations have been introduced to remove SRSF3 binding sites (positions within the transcript denoted by red asterisks in the gene diagram). SRSF3 crosslink nucleotides are shown in red, SRSF3 consensus sequences are highlighted in yellow, and the underlined nucleotides were mutated to abolish SRSF3 consensus binding sites without affecting the coding potential. (**E**) RNA immunoprecipitation (IP) of WT- and ΔSRSF3-*Nanog* mRNA in SRSF3-GFP or GFP-only expressing pluripotent cells. The *Nanog* mRNA enrichment is presented as IP/input and normalised to GFP-only control cells expressing WT-*Nanog* (mean ± SEM, n = 2). (**F**) *Srsf3*-KO (KO) and control (Ctrl) iPSCs expressing WT-*Nanog* or ΔSRSF3-*Nanog* were fractionated into nuclear, cytoplasmic and total fractions 24 hr after 4OHT. Quantification of WT-*Nanog* and ΔSRSF3-*Nanog* mRNA levels in the nuclear and cytoplasmic fractions is presented as nuclear to cytoplasmic ratio and normalised to the control cells expressing WT-*Nanog* (*p<0.05, One-way ANOVA, data as mean ± SEM, n = 3). (**G**) RNA immunoprecipitation of WT- and ΔSRSF3-*Nanog* mRNA in SRSF3-GFP or GFP-only expressing HEK293

*Figure 4 continued on next page*

*Figure 4 continued*

cells. Anti-NXF1 and anti-GFP antibodies were used to enrich mRNAs bound by NXF1 and SRSF3-GFP, respectively. The *Nanog* mRNA enrichment is presented as IP/input and normalised to GFP-only cells expressing WT-*Nanog* (mean ± SEM, n = 2). See also *Figure 4—figure supplement 1*.
DOI: https://doi.org/10.7554/eLife.37419.012

The following figure supplement is available for figure 4:

**Figure supplement 1.** SRSF3 binding is essential for *Nanog* mRNA export.
DOI: https://doi.org/10.7554/eLife.37419.013

that SRSF3 binding is required for the association of NXF1 with *Nanog* mRNA. This is to our knowledge one of the first direct demonstrations of the functional importance of SRSF3 as an mRNA export adaptor as well as of the significance of RNA processing in the control of *Nanog* expression.

## SRSF3 regulated alternative splicing in pluripotent cells and during reprogramming

Given that SR proteins are well-established regulators of pre-RNA splicing (*Änkö, 2014*; *Zhong et al., 2009*), we next performed RNA-sequencing of control and SRSF3-deficient iPSCs to investigate how SRSF3 impacts AS. We quantified the levels of alternatively spliced isoforms with MISO (Mixture-of-Isoforms) probabilistic framework (*Katz et al., 2010*) using the annotated AS events for mouse GRCm38/mm10. Each AS event was assigned a 'percent spliced in' (PSI) or 'percent intron retained' (PIR) value to represent the percent of transcripts with the exon or intron included. Using a cut-off of $\Delta PSI \geq |0.2|$ and Bayes factor $\geq 5$, SRSF3 depletion led to a change in 281 annotated AS events, of which the skipped exon (SE) was the most affected category, both in terms of absolute numbers and proportion of assessed events (*Figure 5A*, *Figure 5—source data 1*). Overall, more exons showed increase in skipping following SRSF3 depletion, in agreement with the role of many SR proteins in enhancing exon inclusion. Sixty-one percent of the altered AS events had associated SRSF3-binding sites within the transcript region based on our iCLIP data. Because skipped exons were the most affected AS type in the SRSF3 deficient cells providing a sufficient number of data points for global analysis, we next mapped SRSF3-bindings sites around these exons. SRSF3 binding was enriched within the regulated exons, with few sites also mapping up- and downstream (*Figure 5B*). Inspection of SRSF3-binding sites around exons decreasing in inclusion in SRSF3 depleted cells suggested that SRSF3 binding within the regulated exon or intron enhances inclusion across the AS types (SE, IR, A5SS and A3SS) which is in accordance with SRSF3-binding sites acting as splicing enhancers (*Figure 5C*).

As regulated splicing plays a central role during reprogramming (*Cieply et al., 2016*; *Ohta et al., 2013*), we next assessed AS changes taking place in the course of reprogramming by performing RNA sequencing of control MEFs, reprogramming intermediates and iPSCs. We first determined cumulative changes in AS during reprogramming using MISO by comparing each of the time points to MEFs, which showed a gradual increase in AS from MEFs to iPSCs (*Figure 5D*, top panel) similar to previous reports (*Cieply et al., 2016*). Sequential comparisons between the time points revealed, however, that AS changes during reprogramming took place in two waves similar to transcriptional changes (*Polo et al., 2012*). The apparent increase in AS towards iPSCs reflected major AS pattern alterations early between MEFs and day 3, and late between day 12 and iPSC (*Figure 5D*, bottom panel). When analysing the different types of AS events in the course of reprogramming, skipped exons were the major type across the time course although all AS types, including intron retention, were dynamically regulated during reprogramming (*Figure 5E* and *Figure 5—figure supplement 1A*, *Figure 5—source data 2* and *3*).

Because MISO only assesses a limited set of previously annotated AS events, we also analysed regulated RNA processing during reprogramming using JunctionSeq which has been reported to readily detect novel splice junctions and intron retention (*Figure 6—source data 1–2*) (*Hartley and Mullikin, 2016*). Principal component analysis showed a molecular connectivity reflecting the progression from MEFs to iPSCs (*Figure 6A*), reminiscent but independent of the connectivity at the transcriptome level (*Figure 6—figure supplement 1A–B*) (*Polo et al., 2012*). Similar to the analysis of annotated AS events using MISO, the number of regulated RNA features identified by Junction-Seq increased from MEFs to iPSC when each of the reprogramming intermediates was compared to

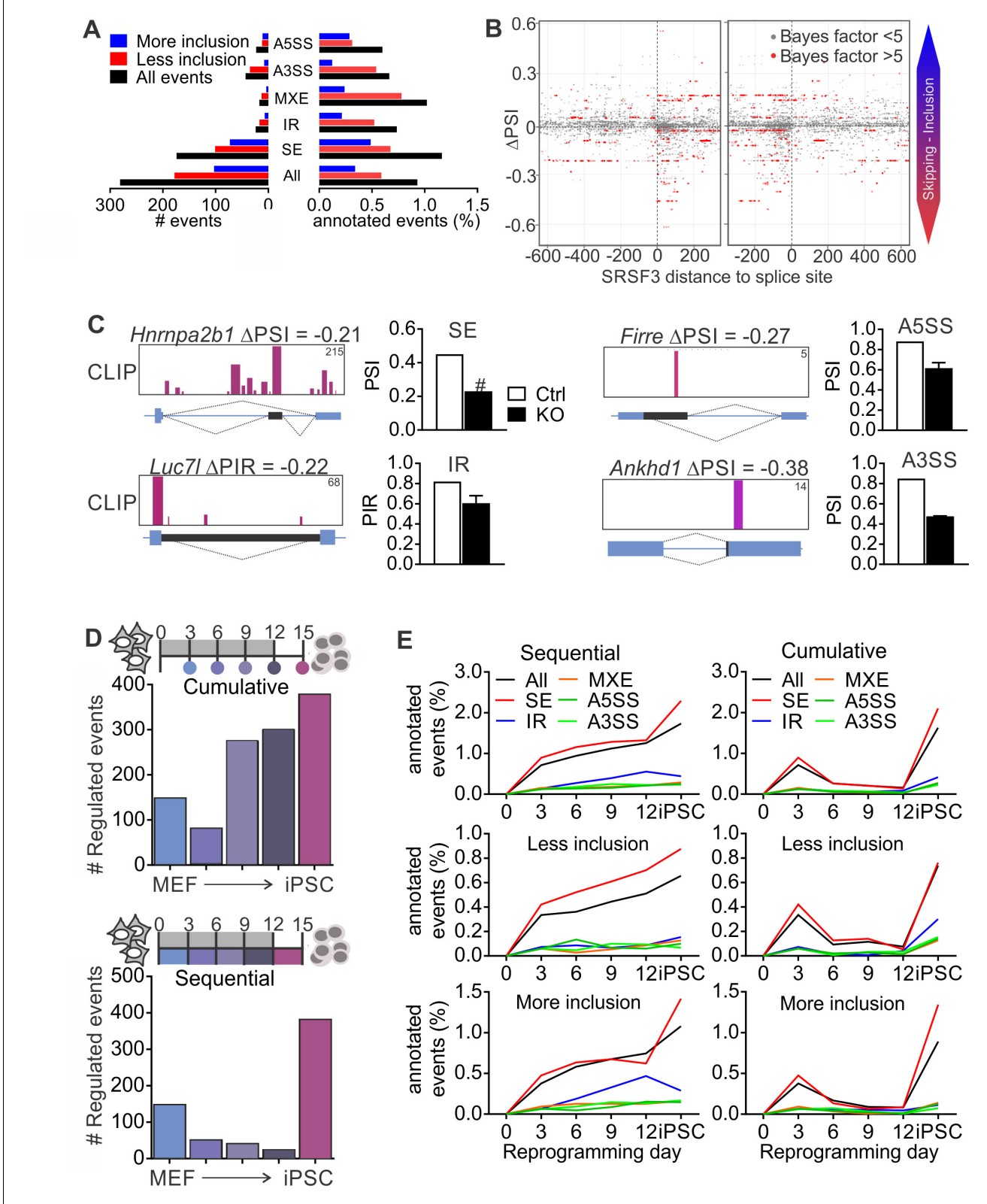

**Figure 5.** SRSF3 regulated alternative splicing in pluripotent cells and the dynamics of AS during reprogramming. (**A**) Different alternative splicing (AS) types affected by SRSF3 depletion in pluripotent cells. Pluripotent cells were subjected to RNA sequencing 24 hr following SRSF3 depletion and AS changes were quantified by MISO (ΔPSI ≥ |0.2| and Bayes factor ≥5). SE, skipped exon; IR, intron retention; MXE, mutually exclusive exon; A3SS, alternative 3' slice site; A5SS, alternative 5' splice site. (**B**) SRSF3 binding sites (iCLIP) at and around SE events that were annotated by MISO. The red

*Figure 5 continued on next page*

*Figure 5 continued*

dots represent AS events that were considered significantly changed following SRSF3 depletion (Bayes factor >5) and grey dots events that did not change (Bayes factor <5). The opacity of the dots corresponds to the $\log_2$(iCLIP count) in the range 1 to 5 before log transformation. Significant SRSF3-binding sites with iCLIP count $\geq$2 that were not detected in the control iCLIP are shown. The dotted lines at 0 denote the 3' and 5' splice sites. (C) Representative examples of different AS types affected by SRSF3 depletion. SRSF3-binding sites mapping at and around the affected event are shown over the gene diagram (PSI values based on MISO, data as mean ± SEM, n = 2). PSI, percent spliced in; PIR, percent intron retained. Ctrl, control cells; KO, SRSF3-depleted cells. (D) Dynamic AS changes take place during reprogramming (MISO, ΔPSI $\geq$ |0.2| and Bayes factor $\geq$5). Comparison of each time point to MEFs showing a cumulative increase in AS from MEFs to iPSC (*top*) and assessment of consecutive time points demonstrating that the majority of AS changes take place in only two time points (*below*). (E) Different AS types display distinct dynamics during reprogramming. Cumulative and sequential change as in D, event types as in A. See also *Figure 5—figure supplement 1* and *Figure 5—source data 1–3*.

DOI: https://doi.org/10.7554/eLife.37419.014

The following source data and figure supplement are available for figure 5:

**Source data 1.** Alternative splicing events following SRSF3 depletion.
DOI: https://doi.org/10.7554/eLife.37419.016
**Source data 2.** Alternative splicing events during reprogramming relative to MEFs.
DOI: https://doi.org/10.7554/eLife.37419.017
**Source data 3.** Alternative splicing events between consecutive time points during reprogramming.
DOI: https://doi.org/10.7554/eLife.37419.018
**Figure supplement 1.** Alternative splicing during reprogramming.
DOI: https://doi.org/10.7554/eLife.37419.015

MEFs (*Figure 6—figure supplement 1C*). Sequential comparisons between each consecutive time point further demonstrated that the majority of RNA processing changes occurred at only two time points (*Figure 6—figure supplement 1C*), coinciding with the biphasic increase in *Srsf3* expression (see *Figure 1B*). We next mapped SRSF3-binding sites within and adjacent to the regulated RNA features identified by JunctionSeq (−200, +200 nt of the junction) to identify 'SRSF3 associated regulated RNA features' or 'SARFs'. Unsupervised k-means clustering grouped 53% of SARFs within 3 out of 20 clusters, each showing a similar upwards trajectory towards iPSCs (*Figure 6B*). As these three clusters represented only 33% of all regulated RNA features, we did not just detect the most prominent clusters but likely captured SRSF3 associated AS events relevant to the pluripotent cell fate. This examination shows that SRSF3 binding is associated with splicing changes contributing to the commitment to the pluripotent cell fate during reprogramming.

Examination of the SARFs during reprogramming revealed surprisingly abundant SRSF3 binding within intronic sequences (*Figure 6C,i*). A more detailed inspection detected a significant proportion of intronic SRSF3-binding sites mapping around so-called detained introns (DIs) (*Figure 6C,ii,iii*). DIs are defined as introns spliced out at a slower rate than an average intron, thus representing a rate-limiting regulatory step during splicing (*Boutz et al., 2015*). Only a small fraction of DIs were annotated as IR events and assessed by MISO but mapping of DIs to our JunctionSeq data identified differential DI inclusion from MEFs to iPSCs, with an increase in DI inclusion towards iPSCs (*Figure 6—figure supplement 1D*). SRSF3-binding sites were found in 37% of regulated DIs in contrast to only 27% of non-regulated DIs (*Figure 6C,iii*). Similar to SRSF3 target RNAs (see *Figure 3—figure supplement 1D*), DI enrichment was reported in genes encoding proteins involved in RNA metabolism (*Boutz et al., 2015*). Indeed, SRSF3-binding sites were detected within DI regions of RNA regulators such as *Nxf1* pre-mRNA (*Figure 6D*). SRSF3-binding sites mapped to *Nxf1* intron 10, which was classified as DI and dynamically regulated during reprogramming (*Figure 6D–E* and *Figure 6—figure supplement 1E*). Although the level of intron containing *Nxf1* mRNA was similar in MEFs and iPSCs, up to 40% intron retention was detected in the reprogramming intermediates (*Figure 6E*). SRSF3 ablation led to a decrease in the level of intron 10 containing *Nxf1* mRNA (*Figure 6F*), suggesting that SRSF3 binding is required for the dynamic changes in *Nxf1*-Int10 retention during reprogramming. The intron 10 containing *Nxf1* transcript encodes a short isoform of NXF1 protein that can bind to RNA but cannot interact with the nuclear pore complex (*Li et al., 2016*). NXF1 has been shown to bind to this intron to allow for the efficient export of the intron containing mRNA (*Li et al., 2006*; *Müller-McNicoll et al., 2016*). Therefore, SRSF3 does not only act as an export adaptor of distinct mRNAs as exemplified by *Nanog* mRNA in pluripotent cells but also it may affect mRNA export more broadly by regulating NXF1 isoform expression and function through *Nxf1* intron 10.

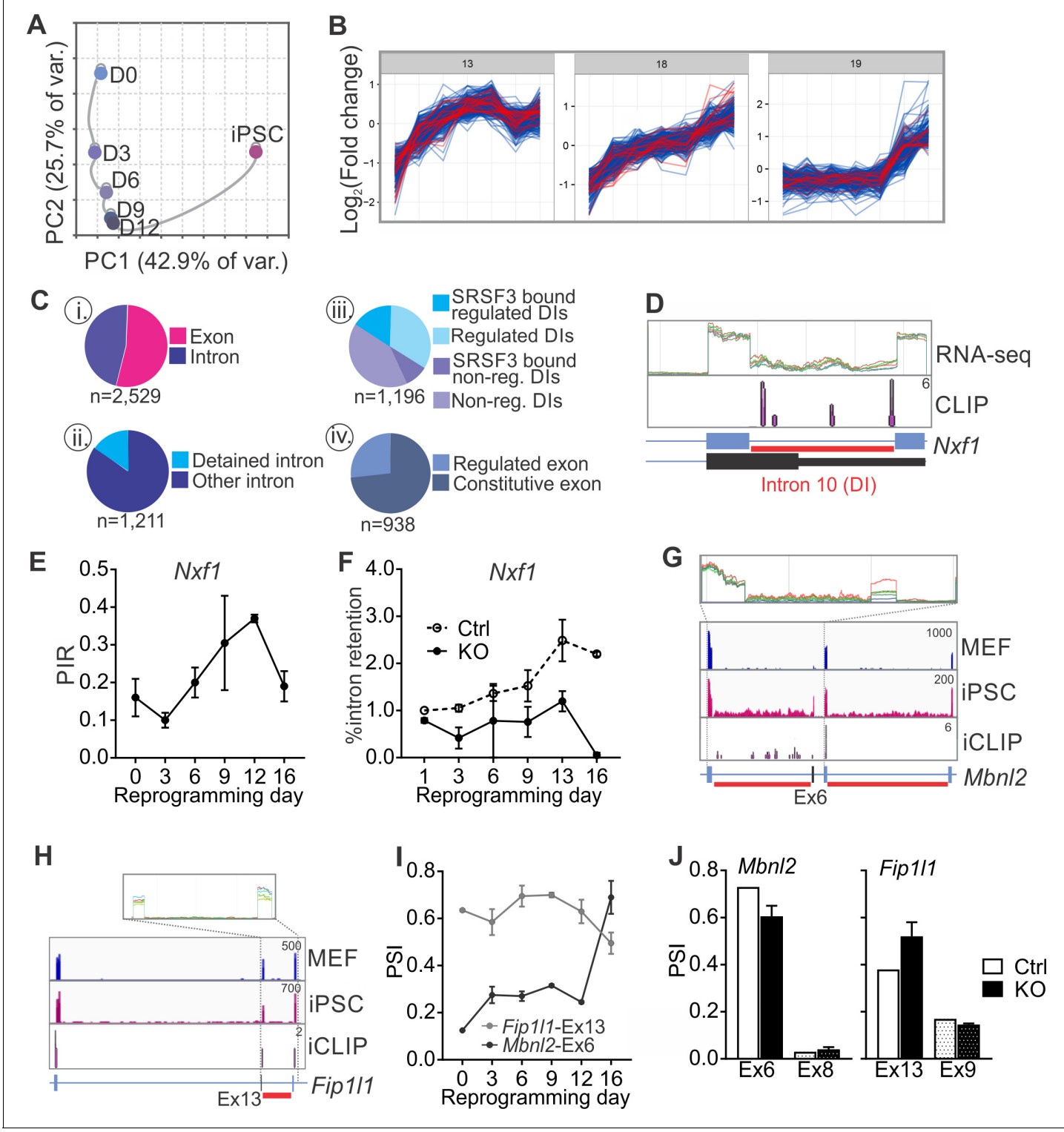

**Figure 6.** SRSF3 regulates the alternative splicing of *Nxf1* and other RNA regulators during reprogramming. (**A**) Principal component analysis of regulated RNA features based on JunctionSeq during reprogramming, depicting similarity of exon/junction usage between reprogramming intermediates. (**B**) Unsupervised k-means clustering of regulated RNA features during the reprogramming time course. SRSF3 associated regulated RNA features, SARFs, are highlighted in red. SRSF3 binding sites within the feature or [−200, +200 nt] window up- or down-stream of the junction were included in the analysis. Top three clusters with most SRSF3-binding sites are shown. (**C**) (i) Distribution of SRSF3-binding sites within regulated exons and introns. Regulated RNA features were classified as exons and introns using RefSeq annotation. (ii) Proportion of intronic SARFs that were classified

*Figure 6 continued on next page*

*Figure 6 continued*

as detained introns (*Boutz et al., 2015*). (iii) Distribution of SRSF3 binding sites within regulated and non-regulated (non-reg.) DIs. (iv) Proportion of SARFs that are DIs and flank an exon regulated during reprogramming. (D) Expression of *Nxf1* Ex10-Int10-Ex11 during reprogramming and SRSF3 iCLIP binding profile within the transcript region. The red bars below the gene diagram denote DIs based on (*Boutz et al., 2015*). Colour coding from blue (MEF) to red (iPSC). The read counts were normalised within the genes. (E) Dynamic changes in *Nxf1*-Int10 retention during reprogramming as measured by RNA-seq (data as mean ± SEM, n = 2). PIR, percent intron retained. (F) RT-qPCR quantification of *Nxf1*-Int10 retention during reprogramming in control (Ctrl) and SRSF3-depleted (KO) cells. The data is represented as percent of transcripts with the retained intron (data as mean ± SEM, n = 2). (G) Expression of *Mbnl2* exons 5–8 and SRSF3 iCLIP binding profile within the transcript region. The red bars below the gene diagram denote DIs based on (*Boutz et al., 2015*). The alternative exon six is highlighted in black in the gene diagram. Colour coding from blue (MEF) to red (iPSC). The read counts were normalised within the genes. (H) Expression of *Fip1l1* exons 12–14 and SRSF3 iCLIP binding profile within the transcript region. The red bars below the gene diagram denote DIs based on (*Boutz et al., 2015*). The alternative exon 13 is highlighted in black in the gene diagram. (I) Dynamic changes in *Mbnl2*-Ex6 and *Fip1l*-Ex13 inclusion during reprogramming as measured by RNA-seq (data as ± SEM, n = 2). PSI, precent spliced in. (J) Quantification of *Mbnl2*-Ex6 and *Fip1l*-Ex13 inclusion in control (Ctrl) and *Srsf3*-KO (KO) iPSCs by RNA-seq (data as mean ± SEM, n = 2). Unaffected alternative exons that were not associated with SRSF3 binding sites (*Mbnl2*- Ex8 and *Fip1l1*-Ex9) are shown for comparison. PSI, percent spliced in. See also *Figure 6—figure supplement 1*, *Figure 6—source data 1* and *2*.

DOI: https://doi.org/10.7554/eLife.37419.019

The following source data and figure supplement are available for figure 6:

**Source data 1.** Read-wide mapping statistics.
DOI: https://doi.org/10.7554/eLife.37419.021
**Source data 2.** Data file containing JunctionSeq output.
DOI: https://doi.org/10.7554/eLife.37419.022
**Figure supplement 1.** SRSF3 controls the alternative splicing of RNA regulators.
DOI: https://doi.org/10.7554/eLife.37419.020

In addition to *Nxf1*, we identified multiple other factors involved in RNA metabolism, including *Mbnl2*, *Fip1l1*, *Clk1* and *Ewsr1* that had SARFs in RNA regions with a DI (*Figure 6G–H* and *Figure 6—figure supplement 1F*). Interestingly, MBNL2 and FIP1 have previously been shown to play regulatory roles in pluripotency (*Han et al., 2013*; *Lackford et al., 2014*) and CLK1 mediates dynamic changes in intron detention in ESCs (*Boutz et al., 2015*). Further assessment of these and other SARFs that were DIs showed a quarter of them flanking alternative exons that were regulated during reprogramming (*Figure 6D, iv*). The inclusion of the AS exons flanking DIs was dynamically regulated during reprogramming (*Figure 6I* and *Figure 6—figure supplement 1G*) and SRSF3 ablation led to changes in their inclusion (*Figure 6J* and *Figure 6—figure supplement 1H*). Of note, the alternative exon in *Srsf3* itself is flanked by DIs and we and others have previously shown that SRSF3 autoregulates the splicing of this exon (*Änkö et al., 2012*; *Lareau et al., 2007*; *Ni et al., 2007*). These data show that the splicing of DIs provide a regulatory step in gene expression during reprogramming and demonstrates a role for SRSF3 in splicing of rate limiting DIs in pluripotent cells.

## SRSF3 regulates the expression of chromatin modifiers essential for the repression of lineage-specific genes

SRSF3 does not only regulate splicing but also global gene expression (*Änkö et al., 2010*). In pluripotent cells, we identified 803 differentially expressed genes (DEGs) following SRSF3 depletion (482 up- and 321 downregulated, FDR < 0.05, log|FC| $\geq$ 1) (*Figure 7A* and *Figure 7—source data 1*). The GO terms associated with the dysregulated genes reflected the loss of pluripotency following SRSF3 depletion (*Figure 7—figure supplement 1A*). Forty-four percent of DEGs had SRSF3 binding sites within transcript region (351 of 802), with no difference between genes that were up- (43%) or downregulated (45%). SRSF3 has been reported to regulate gene expression including its own mRNA abundance via NMD surveillance pathway (*Änkö et al., 2012*; *Lareau et al., 2007*; *Ni et al., 2007*). We compared the SRSF3 CLIP RNA targets to previously identified transcripts regulated by NMD in mouse ESCs (*Hurt et al., 2013*). This revealed that almost half of the NMD-regulated genes/mRNAs had SRSF3-binding sites, suggesting that SRSF3-mediated RNA regulation may be broadly linked to NMD (*Figure 7B*). Further analysis demonstrated that DEGs with SRSF3-binding sites were enriched in the set of genes/mRNAs regulated by NMD (*Figure 7C*). The larger overlap within mRNAs [or 'consistent isoforms' in *Hurt et al. (2013)*] rather than genes suggested that SRSF3 may link distinct transcript variants to NMD.

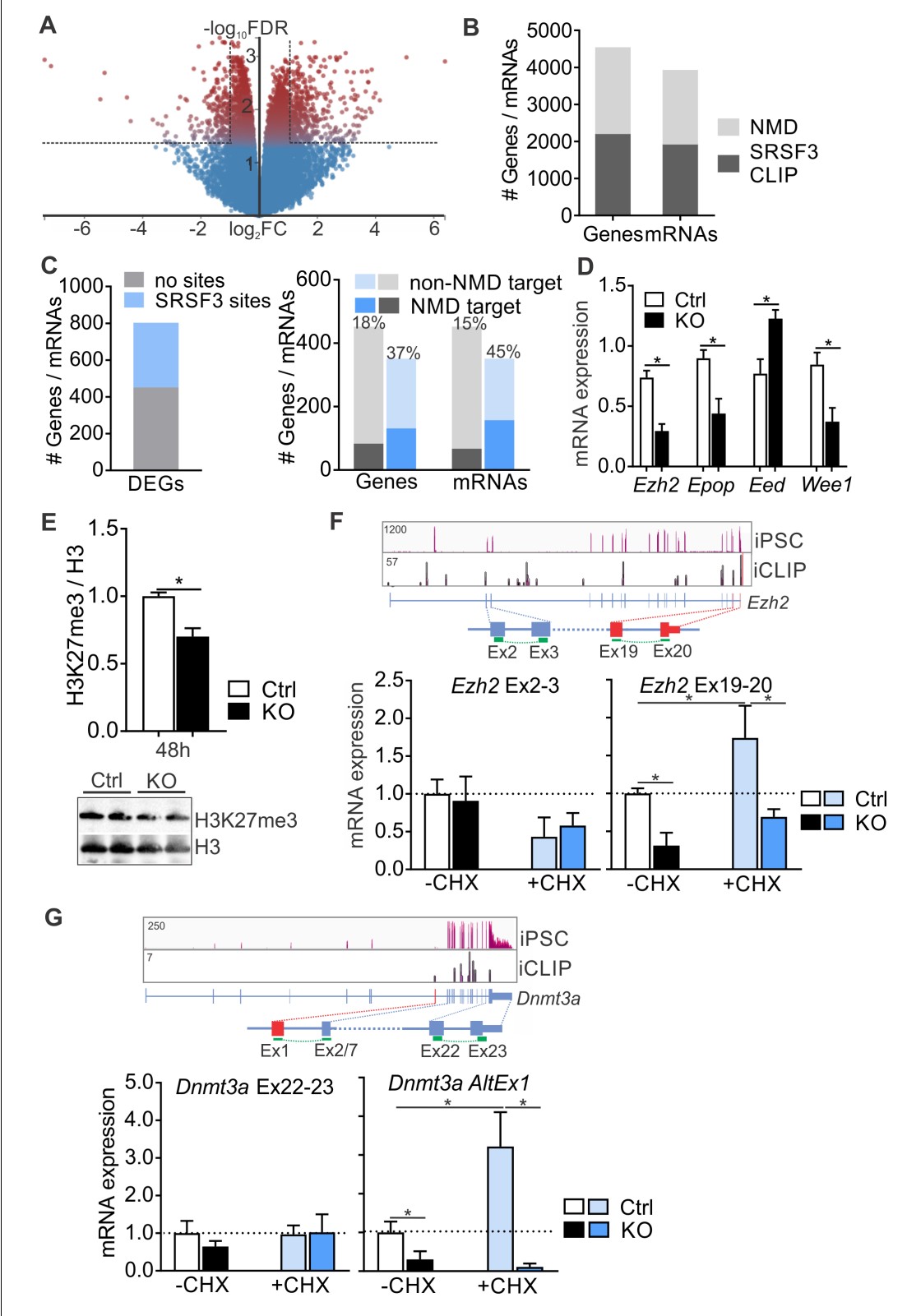

**Figure 7.** SRSF3 mediates the abundance of distinct mRNAs encoding chromatin modifiers to facilitate pluripotency. (**A**) Volcano plot depicting differentially expressed genes following SRSF3 depletion in pluripotent cells. The dotted lines illustrate fold change (FC) >2 and FDR < 0.05. (**B**) Number of genes and mRNAs regulated at the level of NMD based on *Hurt et al. (2013)* and the fraction of them with significant SRSF3 crosslink sites. (**C**) *Left*: Differentially expressed genes (DEGs) following SRSF3 depletion with and without SRSF3-binding sites. *Right*: The fraction of differentially

*Figure 7 continued on next page*

*Figure 7 continued*

expressed genes/mRNAs with (blue bars) and without (grey bars) SRSF3-binding sites that were regulated at the level of NMD. The percent value above the bars denotes the fraction of NMD regulated genes or mRNAs (dark blue or grey) in each category. (**D**) RT-qPCR quantification of PRC2 components in *Srsf3*-KO (KO) and control (Ctrl) iPSCs 24 hr after 4OHT induction. The data is normalised to *Hprt* and presented relative to control iPSCs (*p<0.05, Unpaired Student's t-test, two-tailed, data as mean ± SEM, n = 3–9). (**E**) Western blot analysis of H3K27me3 and total H3 levels 48 hr after 4OHT induction in *Srsf3*-KO (KO) and control (Ctrl) iPSCs. Quantification presented as H3K27me3 relative to total H3 (*p<0.05, Unpaired Student's t-test, two-tailed, data as mean ± SEM, n = 2). (**F**) *Top: Ezh2* expression in iPSCs and SRSF3 iCLIP binding profiles within the transcript region. Primers binding to the exons in red were used to measure the NMD-regulated transcript variant. *Below:* RT-qPCR quantification of total *Ezh2* (Ex2-3) and full-length transcript variant (Ex19-20) in control (Ctrl) and *Srsf3*-KO (KO) iPSCs before (-CHX) and after (+CHX) NMD inhibition. The data is normalised to *Hprt* and presented relative to control iPSCs (*p<0.05, One-way ANOVA, data as mean ± SEM, n = 3). (**G**) *Top: Dnmt3a* expression in iPSCs and SRSF3 iCLIP binding profiles within the transcript region. Primers detecting the alternative first exon (red) were used to measure the NMD-regulated transcript variant. *Below:* RT-qPCR quantification of total *Dnmt3a* (Ex22-23) and *Dnmt3a* alternative exon 1 (AltEx1) containing transcripts as in (**F**). See also *Figure 7—source data 1*.

DOI: https://doi.org/10.7554/eLife.37419.023

The following source data and figure supplement are available for figure 7:

**Source data 1.** Differentially expressed genes following SRSF3 depletion.

DOI: https://doi.org/10.7554/eLife.37419.025

**Figure supplement 1.** SRSF3 controls the abundance of chromatin modifiers.

DOI: https://doi.org/10.7554/eLife.37419.024

SRSF3 RNA targets were enriched in genes encoding chromatin modifiers including components of the PRC2 complex that in pluripotent cells deposits repressive H3K27me3 histone marks to suppress differentiation (*Zhang et al., 2011*) and *Dnmt3a* that encodes DNA methyl transferase 3A mediating gene silencing through the transfer of methyl groups to CpG islands (*Figure 3D* and *Figure 3—figure supplement 1D*). Interestingly, many of these mRNAs were also identified as NMD targets and have been shown to have short half-lives in pluripotent cells (*Hurt et al., 2013*; *Ke et al., 2017*). The expression of multiple components of PRC2 was altered in SRSF3-deficient cells (*Figure 7D*) and consistent with this, the H3K27me3 levels were globally reduced (*Figure 7E*). *Ezh2* encoding the histone-lysine methyltransferase of PRC2 was among the NMD-sensitive SRSF3-bound mRNAs. Based on Ensembl gene annotation and previous studies (*Shirahata-Adachi et al., 2017*), *Ezh2* produces at least three transcript variants, one of which terminates at exon 7. Only the full-length (Ex1-20) variant was prominently expressed in pluripotent cells and SRSF3 binding sites were deposited along this mRNA (*Figure 7F*). To investigate the involvement of SRSF3 and NMD in the control of *Ezh2* expression, we measured *Ezh2* transcript variant mRNA levels in iPSCs treated with CHX. *Srsf3* mRNA, a known NMD target, served as a control for successful NMD inhibition (*Figure 7—figure supplement 1B*). The total level of *Ezh2* determined by primers spanning exons 2–3 (Ex2-3) was not affected by NMD inhibition, whereas the level of the full-length *Ezh2* measured by primers spanning Ex19-20 was increased (*Figure 7F*). Intriguingly, the full-length *Ezh2* mRNA was downregulated in *Srsf3*-KO iPSCs (*Figure 7F*), providing evidence that SRSF3 specifically regulates the NMD-sensitive *Ezh2* variant. NMD inhibition increased and SRSF3 depletion decreased also the level of the intronless *Epop* mRNA encoding an ESC specific component of PRC2 (*Figure 7—figure supplement 1C*), suggesting that SRSF3 controls the expression of multiple components of the PRC2 complex. Similar to *Ezh2*, SRSF3 bound to and regulated the expression of a distinct NMD-sensitive variant of *Dnmt3a* (*Dnmt3a*-AltEx1) (*Figure 7G*). Interestingly, none of the investigated transcript variants contained premature termination codons (PTCs) generally associated with mRNAs targeted to NMD. To dissect if SRSF3 affected the production or stability of the NMD-sensitive transcript variants, we determined mRNA half-lives in Actinomycin D treated control and SRSF3-depleted pluripotent cells. *Myc* mRNA served as a control showing the expected half-life of 0.7 hr in ESCs (*Ke et al., 2017*). The half-lives of the NMD-sensitive mRNAs were not significantly altered following SRSF3 depletion (*Figure 7—figure supplement 1D*), suggesting that SRSF3 regulates the generation of distinct mRNAs whose steady-state levels are maintained by NMD to balance production and turnover. Taken together, SRSF3 regulates the abundance of chromatin modifiers through binding to distinct NMD-sensitive transcript variants, SRSF3 depletion leading to global changes in the chromatin state of pluripotent stem cells.

## Discussion

Here, we elucidate the essential role of SRSF3 in reprogramming and self-renewal by directly regulating key components of the NANOG-centred pluripotency circuitry (*Figure 8*). SRSF3 activity in coordinating the pluripotency gene expression program illustrates the central role of RBPs in modifying transcriptomes and regulating cell fate decisions.

The importance of RNA processing as a means to regulate cell fate has only recently started emerging (*Cieply et al., 2016*; *Han et al., 2013*; *Lackford et al., 2014*). The functions of SR proteins, including SRSF3, have expanded from essential regulators of splicing to multifaceted mediators of RNA processing (*Zhong et al., 2009*). High-throughput technologies have allowed the cataloguing of SR protein targets, but the impact of SR protein RNA binding at the cellular level has remained largely elusive. Using our reprogrammable *Srsf3* knockout model that allowed a systematic investigation of SRSF3 activity in the generation and maintenance of iPSCs, we demonstrate that SRSF3 is required for both the proliferative phase and the later commitment to pluripotency. The proliferative response of the first wave of reprogramming was largely lost in SRSF3 deficient cells, in accordance with the previous RNAi screen (*Ohta et al., 2013*). However, SRSF3 depletion late during reprogramming also greatly reduced the generation of AP-positive pluripotent colonies. We show that SRSF3 controls the commitment to pluripotency directly by targeting key genes of the pluripotency network. Furthermore, overexpression of SRSF3 led to an increase in AP/SSEA1-positive cells, suggesting that SRSF3 could be used to enhance reprogramming efficiency.

Combination of iCLIP and RNA-seq revealed the significance of SRSF3-mediated RNA regulation for the establishment and maintenance of pluripotency. Remarkably, SRSF3 controlled the expression of the core pluripotency transcription factor NANOG by facilitating the nucleo-cytoplasmic export of *Nanog* mRNA. The nuclear sequestration of *Nanog* mRNA in SRSF3-deficient iPSCs led to greatly reduced NANOG protein levels. Consistent with this, the cellular phenotype of SRSF3-depleted pluripotent cells resembled that of *Nanog*-null cells (*Mitsui et al., 2003*). We demonstrate that the function of SRSF3 in mRNA export is independent of splicing by showing that ectopically expressed intronless wild-type *Nanog* mRNA is efficiently exported. Mutation of SRSF3 consensus binding sites abolished SRSF3 binding to *Nanog* mRNA and resulted in the nuclear accumulation of the mRNA. In the absence of SRSF3-binding sites, the interaction between *Nanog* mRNA and the

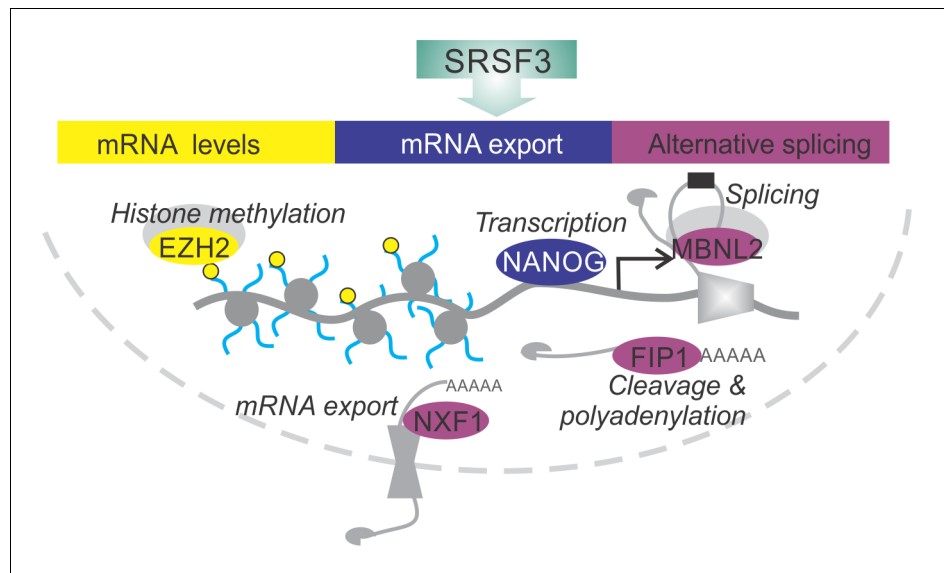

**Figure 8.** SRSF3 drives self-renewal by regulating RNAs encoding key pluripotency factors. SRSF3 directly regulates the core pluripotency network through its activities in mRNA export, alternative splicing and mRNA turnover. Through the control of RNAs encoding chromatin modifiers, transcription and RNA processing factors that are involved in the establishment and maintenance of pluripotency SRSF3 has a broad impact on the pluripotency gene expression program.
DOI: https://doi.org/10.7554/eLife.37419.026

export factor NXF1 was lost, suggesting that SRSF3 is necessary and sufficient for the recruitment of NXF1 to *Nanog* mRNA. Interestingly, knockdown of THO complex components in ESCs resulted in mRNA export defects and loss of pluripotency (*Wang et al., 2013*), suggesting that RBP-regulated export of distinct mRNAs may be broadly used to control gene expression in pluripotent cells.

SRSF3 directly regulated the splicing of multiple pre-mRNAs encoding established pluripotency factors, including other RNA regulators. In particular, SRSF3 bound to pre-mRNAs containing post-transcriptionally spliced detained introns (*Boutz et al., 2015*), one such pre-mRNA encoding the export receptor NXF1. We demonstrate that during reprogramming intron detention is dynamically regulated and may significantly modulate gene expression output in cell fate transitions. SRSF3 bound to and regulated the inclusion of DIs, as exemplified by *Nxf1* pre-mRNA. This extends the well-established role of SRSF3 in enhancing exon inclusion by binding to exonic splicing enhancers to intronic enhancer sequences. SRSF3 increased the production of an intron containing *Nxf1* transcript variant that encodes a short NXF1 isoform incapable of interacting with the nucleopore complex (*Li et al., 2016*). Thus, SRSF3 interacts with NXF1 at multiple levels, firstly by regulating *Nxf1* splicing and secondly, by functioning as an mRNA export adaptor of NXF1 at the protein level. This suggests that SRSF3 can influence mRNA export both by regulating NXF1 isoform expression and by directly mediating the export of distinct mRNAs such as *Nanog* in pluripotent cells. More broadly, our data identifies SRSF3 as a regulator of DI-containing pre-mRNAs encoding RBPs beyond *Nxf1*, suggesting that SRSF3 may influence RNA processing in pluripotent cells not only by regulating its direct RNA targets but also through the regulation of other RBPs (*Figure 8*).

Finally, SRSF3 depletion in pluripotent cells led to the downregulation of hundreds of mRNAs, many of which were direct SRSF3 targets. Interestingly, these mRNAs were also regulated at the level of NMD in pluripotent cells (*Hurt et al., 2013*). SRSF3 depletion affected explicitly the level of the NMD-sensitive transcript variants, suggesting a link between SRSF3 and mRNA surveillance pathway. One functional class of NMD-sensitive SRSF3 target mRNAs was chromatin modifiers involved in resetting the epigenome during reprogramming. As a functional output SRSF3 depletion led to a global decrease in repressive H3K27me3 histone marks that are deposited on the promoters of lineage regulators to suppress differentiation in pluripotent cells (*Zhang et al., 2011*). Surprisingly, the NMD-sensitive transcript variants did not contain PTCs that typically mark mRNAs for degradation. Recent studies have demonstrated that besides degrading aberrant PTC-containing mRNAs, NMD targets many seemingly 'normal' mRNAs (*Colombo et al., 2017*; *Imamachi et al., 2017*) (*Hurt et al., 2013*). The mechanism by which normal mRNAs are targeted to NMD and the exact role of SRSF3 in the process will require further investigation.

Our results reveal the multifaceted role of SRSF3 in the control of gene expression programs during reprogramming and in self-renewing cells. The determination of SRSF3-mediated RNA processing in pluripotent stem cells not only highlights the significance of post-transcriptional gene regulation during self-renewal but also provides key mechanistic insights into SRSF3 functions during early development. *Srsf3*-null mice die before the blastocysts stage (*Jumaa et al., 1999*), the source of ESCs in the mouse embryo. It is tempting to propose that the mechanisms identified here are applicable to early development. Perhaps *Srsf3*-null embryos cannot establish the pluripotency program due to premature activation of lineage-specific genes following reduced H3K27me3 levels. With drastically decreased production of NANOG protein, pluripotency would not develop in mice, ultimately causing early embryonic lethality (*Mitsui et al., 2003*; *Silva et al., 2009*). Furthermore, *Srsf3* is a proto-oncogene and its expression is frequently dysregulated in cancer cells (*Jia et al., 2010*; *Kurokawa et al., 2014*). The mechanisms identified here may help elucidating the role of SRSF3 in malignant transformation as cancer (stem) cells share many features with self-renewing cells. In conclusion, our examination highlights the importance of RBPs in coordinating the gene expression program that is required to establish and maintain the pluripotent cell fate.

## Materials and methods

### Mouse strains, primary cells and cell lines

*Srsf3* conditional allele was generated from *Srsf3*tm1a(KOMP)Mbp knockout-first mice (Knock Out Mouse Project, KOMP, UC Davis Mouse Biology Program) via Flpe recombination. These mice were crossed with Tg(UBC-cre/ERT2)1Ejb/1J mice (*Ruzankina et al., 2007*) and further with m3-OKSM mice

(*Alaei et al., 2016*). The resulting reprogrammable *Srsf3* knockout mouse strain (aka *Srsf3*-KO/ OKSM) carried loxP sites flanking exons 2 and 3 of *Srsf3* gene, tamoxifen inducible CreERT2 under the control of *Ubc* promoter, a constitutively expressed CAG-*M3rtTA* reverse transactivator and a multicistronic *OKSM* cassette with a doxycycline (Dox) inducible promoter at the *Col1a1* locus. *Srsf3*-KO/OKSM mice were time mated and embryonic day 13.5 (E13.5) embryos were used for MEF isolation. MEFs isolated from m3-OKSM mice carrying an *Oct4*-GFP reporter (*Alaei et al., 2016*) were used to generate samples for RNA sequencing. All animal works were performed according to the Australian code for the care and use of animals for scientific purposes (NHMRC) and approved by the Monash University Animal Ethics Committee.

SRSF3-BAC ES cells and EGFP-NLS control cells were generated as previously by transfecting BAC-transgene encoding mouse *Srsf3* C-terminally tagged with EGFP or EGFP-NLS vector into mouse JM8A3.N1 ES cells (*Ankö et al., 2010*). The verified JM8A3.N1 ESC cells were obtained from the European Conditional Mouse Mutagenesis Program (EuCOMM) and were mycoplasma negative. Transfected cells were selected using geneticin, followed by enrichment for EGFP-positive cells by FASC. The mycoplasma-negative HEK293T cells were obtained from ATCC (CRL-3216).

## Generation of expression constructs

Lentiviral SRSF3 overexpression construct was generated by subcloning the cDNA encoding human *SRSF3* followed by *T2A* self-cleaving peptide and *EGFP* (*SRSF3-T2A-EGFP*) into the lentiviral vector pCCL. A vector carrying only EGFP was generated and used as a control. WT-*Nanog* and ΔSRSF3-*Nanog* cDNA sequences were ordered as synthetic gene fragments (Biomatik, Canada) and cloned under the EF1alpha promoter using NEBuilder HiFi DNA Assembly (New England Biolabs, MA, USA).

## Reprogramming, ESC/iPSC culture and HEK293 cells

MEFs (passage 1–2) were seeded at the density of 2000 cells/cm$^2$ on collagen coated dishes (High-Glucose DMEM, 15% FBS, 1 mM Sodium Pyruvate, 1 mM MEM non-essential amino acids, 10,000 U/ ml penicillin/Streptomycin and 0.55 mM beta-mercaptoethanol) and cultured at 37°C and 5% CO$_2$/ 5% O$_2$. SRSF3 depletion was induced by 5 uM 4-hydroxy-tamoxifen (4-OHT). OKSM expression was induced by adding doxycycline (Dox, 2 ug/ml) for 13 days in iPSC culture medium (KnockOut DMEM, 15% FBS, 2 mM Glutamax, 1 mM MEM non-essential amino acids, 10,000 U/ml of penicillin/ streptomycin, 1000 U/ml Leukemia Inhibitory Factor (LIF), 1 uM beta-mercaptoethanol), followed by 3 days of culture without Dox. For SRSF3 overexpression, reprogrammable MEFs were seeded as above and spin-infected with a lentiviral vector carrying *Srsf3*-T2A-EGFP or EGFP. Reprogramming was induced 24 hr post-infection with Dox. ESCs and iPSCs were maintained on irradiated MEF feeder cells in iPSC medium without penicillin/streptomycin. MEFs were depleted before iPSC collection for analysis. HEK293 cells were maintained in standard culture media (high-glucose DMEM, 10% FBS, 2 mM Glutamax, 10,000 U/ml of penicillin/streptomycin) at 37°C and 10% CO$_2$.

## AP staining and flow cytometry

Reprogramming efficiency was determined by AP staining (Vector Red Alkaline Phosphatase Substrate Kit, Vector Laboratories, CA, USA) at days 13 and 16. For flow cytometric analysis of cell surface marker expression, the cells were stained with anti-Thy1.2–450 (eBioscience, CA, USA) and anti-SSEA1 (Cell Signalling Technology, MA, USA). Dead cells were excluded by staining with Propidium iodide (PI). Apoptosis was quantified by the proportion of AnnexinV-positive cells (FITC-AnnexinV, BD Pharmingen, NJ, USA), followed by flow cytometric analysis with the BD LSR II flow cytometer. Resulting data were analysed with the FlowJo software.

## Teratoma formation assay

iPSC colonies were picked at day 16 and expanded for 4–5 passages. The iPSCs were MEF depleted and $1 \times 10^6$ cells were injected subcutaneously into the dorsal flanks of immunodeficient 5- to 8-week-old female NOD-SCID mice. After 5 weeks, teratomas were harvested and fixed in 4% paraformaldehyde, embedded in paraffin, sectioned at 5 µm and stained with haematoxylin and eosin.

## Nucleo-cytoplasmic fractionation

Nucleo-cytoplasmic fractionation of control and *Srsf3*-KO iPSCs induced with 4OHT for 24 hr was performed as described (*Müller-McNicoll et al., 2016*) and RNA was isolated as below. The *Nanog* expression constructs were transfected into control or *Srsf3*-KO iPSCs using Lipofectamine 2000 (Thermo Fisher, MA, USA) and 24 hr later 4OHT was added. The fractionation was performed 24 hr post-induction.

## RNA immunoprecipitation

SRSF3-BAC ES cells and EGFP-NLS control cells were transfected with WT-*Nanog* or ΔSRSF3-*Nanog* vectors as above and 24 hr later the cells were MEF depleted and harvested. The cell pellets (1–5 × $10^6$ cells) were suspended into NET-2 buffer (50 mM Tris-HCl pH 7.5, 150 mM NaCl and 0.05% Nonidet P-40). The lysates were sonicated using Bioruptor (Diagenode, NJ, USA) and the supernatants used for RNA immunoprecipitation with anti-EGFP (Abcam, UK, ChIP grade) coupled Dynabeads Protein G (Thermo Fisher). RNA was isolated from the beads and input samples (5% of the lysate) as below for RT-qPCR. Similarly, the *Nanog* constructs were transfected into HEK293 cells using Lipofectamine 2000 and RNA immunoprecipitation was performed 24 hr later. To determine SRSF3 binding to WT-*Nanog* or ΔSRSF3-*Nanog* in HEK293 cells, the *Nanog* vectors were co-transfected with a vector carrying SRSF3-YFP or EGFP-NLS (*Ankö et al., 2010*) and RNA immunoprecipitations were performed as above. The RNA immunoprecipitations using anti-NXF1 antibody (Sigma Aldrich, MO, USA) and HEK293 cell lysates transfected with the WT-*Nanog* or ΔSRSF3-*Nanog* vectors were performed as described above.

## Apoptosis inhibition, NMD inhibition and mRNA stability assays

To inhibit apoptosis, control and *Srsf3*-KO iPSCs cells were treated with 25 uM QVD-OPh with or without 4OHT for 24–48 hr. The cells were MEF depleted and used for RNA or protein isolation or for FACS analysis. To inhibit NMD, control and *Srsf3*-KO iPSCs cells were treated with 4OHT for 24 hr, followed by 100 ug/ml CHX for two hours as in (*Hurt et al., 2013*) before MEF depletion and harvesting and RNA isolation. The RNA was used either for RNA-sequencing or for cDNA synthesis followed by qPCR as described below. For the mRNA stability assays, the 4OHT-induced control and *Srsf3*-KO iPSCs cells were treated with 10 uM actinomycin D for 0–24 hr. Cells were MEF depleted and harvested at the time points indicated in *Figure 7—figure supplement 1*, followed by RNA isolation and RT-qPCR as described below.

## RNA extraction and quantitative real-time PCR

Total RNA was isolated using the TRI Reagent (Sigma-Aldrich) and subjected to DNaseI treatment. RNA was used for cDNA synthesis with SuperScript III Reverse Transcriptase (Thermo Fisher). Quantitative real-time PCR was performed in an AB 7500 real-time PCR system (Applied Biosystems, CA, USA) with Luminaris HiGreen qPCR Master Mix-low ROX (Thermo Fisher) and 0.3 uM primer mix (*Table 1*). The cycle threshold value (CT) was calculated for each sample and normalised to *Hprt*. The relative mRNA levels were calculated using ΔΔCT method.

## Western blotting

For all Western blots, cells were lysed in RIPA buffer (50 mM Tris-HCl pH 8, 150 mM NaCl, 1% Nonidet P40, 0.5% Sodium deoxycholate, 0.1% SDS). NuPAGE 4–12% gradient Bis-Tris gel system (Thermo Fisher) was used in reducing conditions. The nitrocellulose membranes were probed with antibodies against SRSF3 (Sigma-Aldrich), NANOG (Abcam), H3K27me3 (Merck Millipore, Germany), Histone H3 (Abcam), γH2ax (Genetex, CA, USA) and beta-ACTIN (Abcam), followed by HRP-conjugated secondary antibodies (Biorad, CA, USA). The blots were developed using Amersham ECL Western Blotting Detection Reagents (GE Healthcare, UK) and visualised with the Biorad ChemiDoc MP Imaging System (UK). The band intensities were quantified with Fizi (*Schindelin et al., 2012*).

## iCLIP

SRSF3-BAC and EGFP-NLS control ESCs were UV crosslinked with 100mJ/cm$^2$ and iCLIP was performed as in (*Huppertz et al., 2014*). Protein G Dynabeads coupled with goat anti-GFP antibody (D. Drechsel, Max Planck Institute of Molecular Cell Biology and Genetics, Dresden) were used for

**Table 1.** Primers used in this study.
F = forward primer, R = reverse primer.

| Gene symbol | Sequence | |
| --- | --- | --- |
| Srsf3 | F | TGAATTAGAACGGGCTTTTGG |
| | R | TTCACCATTCGACAGTTCCAC |
| Hprt | F | TGTTGTTGGATATGC |
| | R | TGCGCTCATCTTAGG |
| Nanog | F | ACCAAAGGATGAAGTGCAAGC |
| | R | TGGATGCTGGGATACTCCACT |
| Nanog TR | F | GCAGTTTTTCATCCCGAGAAC |
| | R | GAAGAGGCAGGTCTTCAGAGG |
| Nanog FL | F | TGACATGAGTGTGGGTCTTCC |
| | R | GAAGAGGCAGGTCTTCAGAGG |
| Pou5f1 | F | GAGGAAGCCGACAACAATGAG |
| | R | 5'ATCTGCTGTAGGGAGGGCTTC |
| Sox2 | F | GTAAGATGGCCCAGGAGAACC |
| | R | ATAATCCGGGTGCTCCTTCAT |
| Klf4 | F | GAAAAGAACAGCCACCCACAC |
| | R | CCTGTCACACTTCTGGCACTG |
| Zfp42 | F | AGATTAGCCCCGAGACTGAGG |
| | R | AAGGGAACTCGCTTCCAGAAC |
| Myc | F | CTGTACCTCGTCCGATTCCAC |
| | R | GGTTTGCCTCTTCTCCACAGA |
| Nxf1 | F | TTCTGCCTGTCTGTTGTCTCC |
| | R | CAGAACAGAAAAGGGGAGGTG |
| Mbnl2 | F | AAAGCACTGAAGCGACCTCTC |
| | R | AGAGCCTGCTGGTAGTGCAAG |
| Ewsr1 | F | GCTTCAATAAGCCTGGTGGAC |
| | R | TGCCAGATCATCCAGAGTCAC |
| Fil1l1 | F | TCCAATAACTGTACCACCTCCA |
| | R | CCATAGGGAACGCTCGTG |
| Ezh2 Ex2-3 | F | AATCTGAGAAGGGACCGGTTT |
| | R | ATGTGCACAGGCTGTATCCTC |
| Ezh2 Ex19-20 | F | GGGCTATCCAGACTGGTGAAG |
| | R | CCTGAAGCTAAGGCAGCTGTT |
| Epop | F | CCGGCTGATGCTCTTTCTACT |
| | R | CCGCTAAACTGACCCTCATTC |
| Eed | F | GGCAAACTGTATGTTTGGGATT |
| | R | TCGCAGACAGCTATGAGGATG |
| Wee1 | F | GAGCTGGTGAAGCATTCAGTG |
| | R | CATCCGATCTGTGAAGAGTGC |
| Dnmt3a Ex22-23 | F | GGGGACCCCTACTACATCAGC |
| | R | AGAGGCCTGGTTCTCTTCCAC |
| Dnmt3a AltEx1 | F | CCAGACGGGCAGCTATTTACA |
| | R | AGAGGCCTGGTTCTCTTCCAC |
| Nanog-WT and Nanog-ΔSRSF3 | F | CAAGCCTCAGACAGTGGTTCA |
| | R | ATGTCAGTGTGATGGCGAGG |

immunopurification. Cross-linked, immunopurified RNA was digested to lengths of 60–150 nt, reverse-transcribed and subjected to sequencing on an Illumina HiSeq1500.

## RNA-sequencing

For RNA sequencing of reprogramming time course, SSEA1+ reprogramming intermediates at days 3–12 were enriched using AutoMACS Separator (Miltenyi Biotec, Germany) and anti-SSEA1 Microbeads (Miltenyi Biotec) according to the supplier's instructions. The enriched cells were labelled for FACS as in (*Nefzger et al., 2014*). In short, the cells were labelled at approximately $30 \times 10^6$ cells/ml at 4°C. Day 3 and day 6 cells were labelled in a primary labelling step with Anti-Mouse CD90.2 (Thy-1.2) eFluor 450 (eBioscience) and Anti-Human/Mouse SSEA-1 Biotin (eBioscience), followed by a secondary labelling step with APC Streptavidin (BD Pharmingen). Day 9 and day 12 cells were labelled in a primary labelling step with Anti-Mouse CD90.2 (Thy-1.2) eFluor 450 (eBioscience), Anti-Human/Mouse SSEA-1 Biotin (eBioscience) and Anti-Mouse CD117 (c-Kit) APC (eBioscience), followed by a secondary labelling step with Streptavidin PE (eBioscience). Dead cells were excluded by PI. iPSCs were FACS-sorted for Oct4-GFP expression. RNA was extracted from approximately $0.5 \times 10^6$ FACS-sorted cells using the RNeasy Micro Kit (Qiagen, Germany) according to the manufacturer's recommendations. Sequencing libraries were generated using TruSeq RiboZero Kit and Illumina TruSeq protocol, high-throughput sequencing was carried out in duplicates on Illumina HiSeq1500 (100 nt paired end reads, 57–92M reads per sample).

For RNA-sequencing of control and SRSF3 *Srsf3*-KO iPSCs, the cells were induced with 4OHT for 24 hr, MEF depleted and RNA isolated as described above. The sequencing libraries were generated using Illumina TruSeq poly-A selection with strand-specific chemistry. Sequencing was carried out in duplicates on Illumina NextSeq (75nt paired end reads, ~65M reads per sample).

## iCLIP and RNA sequencing data analysis

The iCLIP data was analysed using iCount as described (*Änkö et al., 2012*; *König et al., 2010*; *Müller-McNicoll et al., 2016*; *Wang et al., 2010*). In short, adapters and barcodes were removed from all reads before mapping to the mouse mm10 genome assembly. Uniquely mapping reads were kept to determine statistical significant crosslink sites. For this, iCLIP positions were randomised within co-transcribed regions and statistically significant binding sites (false discovery rate, FDR < 0.05) were calculated. Three biological replicates for SRSF3-BAC and EGFP-NLS control ESCs were pooled for the analysis.

K-mer analysis was performed for motif search as described (*König et al., 2010*; *Wang et al., 2010*). One occurrence of k-mer (4-mers for noncoding RNAs and 5-mers for all RNAs) within the evaluated interval (−30,–5)(+5,+30) around the crosslink site was counted and weighted by 1.0. Reference data was generated 100 times by random shuffling of crosslink sites within corresponding genome segments and z-score was calculated relative to the randomized positions. The consensus binding motif was calculated from top 15 using MEME (*Bailey et al., 2006*). Gene Ontology analysis was performed using DAVID (*Huang et al., 2009*)

Paired-end RNA-seq reads aligned to the mm10 reference genome using STAR (v2.5.2) with default parameters (*Dobin et al., 2013*). The iPS-1 and iPS-2 control replicates were merged using samtools-merge (Samtools v1.2); producing a combined alignment for the control condition. Differential splicing analysis was done using MISO v0.5.4 (LINK1; in a relaxed configuration where filter_results = False) and the mm10 alternative event annotation (version 2; LINK2). For each AS event type, the treatment replicates were compared to the combined control (using miso_compare); this identified putative treatment-sensitive AS events. The results were then filtered using MISO's filter_events command (FILTERS: num-inc 1 = at least one inclusion read, num-exc 1 = at least one exclusion read, num-sum-inc-exc 10 = at least 10 reads of any kind in one of the samples' delta-psi 0.20 = minimum 0.2 delta PSI, bayes-factor 5 = bayes factor at least 5). Similarly, the paired-end MEF-to-iPSC time-course reads were aligned to the mm10 reference genome using STAR (v2.5.2). The differential splicing analysis was done as above.

Splice junctions from the MEF-to-iPSC time-course were reconstituted from the corresponding transcript annotations, flanked by 100 nt up- and downstream of each junction. The resulting

alignments were used as input for JunctionSeq analysis (*Anders et al., 2012*; *Hartley and Mullikin, 2016*), which is based on the DEX-seq model but has been reported to detect readily novel splice junctions and intron retention. Briefly, exon coordinates from transcript models with a Transcript Support Level of 1 (the most stringent) were extracted and used as input for QoRTS (*Hartley and Mullikin, 2015*), together with the alignments from the time-course RNA-seq data. The capability of QoRTS to call novel splice junctions from the data was enabled. The output from QoRTS, a flat exon/junction annotation file, was used together with the alignment data as input for the Junction-Seq software to detect differential exon/junction usage between time-points. Each time-point was contrasted against the original MEF cells (day-0) in one JunctionSeq run. In another run, each time-point was contrasted to the previous time-point. In a third run, all time-points were considered together in the one linear model to test for any effect at all of time-point on exon/junction usage. $Log_2$-fold changes reported are all variance-stabilised transformation of the original $log_2$ fold-changes in expression estimates. In each of the pairwise contrast between timepoints, exonic/junction, features with a FDR < 0.05 were classed as 'regulated', while others were classed either as unregulated (FDR > 0.05), or 'untestable' (too few reads observed in both time points), by JunctionSeq.

Data for SRSF3 iCLIP was filtered for sites that also appeared in the GFP iCLIP control, then converted to bed format. This data was then merged with the exon/junction feature annotations from QoRTS (including de novo junction calls) using BedTools. Detained intron data (*Boutz et al., 2015*) was similarly merged with the QoRTS annotation. For k-means clustering of time-point data, a minimum autocorrelation filter (correlation between consecutive time-points) was set at 0.1, and maximum and minimum $log_2$ fold changes of 8.5 and 0.5, respectively, were also applied. The number of clusters was set to 20. The interaction analysis used for *Figure 3* was performed with STRING (*Szklarczyk et al., 2015*).

## Statistical information

The statistical analysis was conducted using Graph Pad Prism 7 or R. Unpaired Student's t-test was used when two conditions were compared, One-way ANOVA for comparison of three or more experimental conditions and two-way ANOVA when comparisons included multiple variables (such as treatment and time). The number of biological replicates, details of the statistical methods used and how statistical significance was defined is indicated in each figure legend.

## Data availability

RNA-sequencing data for the reprogramming time-course is deposited in GEO Accession Number GSE101905 and RNA-sequencing data for *Srsf3*-KO and control iPSCs in GEO Accession Number is GSE113794. iCLIP data is available in the public version of iCount (http://icount.biolab.si) and as source data to *Figure 3*.

## Acknowledgements

We thank Peter Boag, Jan Kaslin and Kerry Dunse for comments on the manuscript, Jernej Ule for iCount analysis, Jaber Firas for assistance in the teratoma assays, Prof David Huang for providing apoptosis inhibitors and Andrea Aprico for technical support.

## Additional information

### Funding

| Funder | Grant reference number | Author |
| --- | --- | --- |
| National Health and Medical Research Council | GNT1042851 | Traude H Beilharz |
| National Health and Medical Research Council | GNT1092280 | Anja S Knaupp |
| Australian Research Council | | Jose M Polo |

| | |
|---|---|
| Sylvia and Charles Viertel Charitable Foundation | Jose M Polo |
| Jane ja Aatos Erkon Säätiö | Minna-Liisa Anko |
| National Health and Medical Research Council | GNT1043092 | Minna-Liisa Anko |

The funders had no role in study design, data collection and interpretation, or the decision to submit the work for publication.

## Author contributions

Madara Ratnadiwakara, Conceptualization, Data curation, Formal analysis, Validation, Investigation, Visualization, Methodology, Writing—original draft, Writing—review and editing; Stuart K Archer, Data curation, Software, Formal analysis, Visualization, Methodology, Writing—original draft, Writing—review and editing; Craig I Dent, Data curation, Software, Formal analysis, Visualization, Methodology; Igor Ruiz De Los Mozos, Data curation, Software, Formal analysis, Methodology; Traude H Beilharz, Anja S Knaupp, Christian M Nefzger, Investigation, Methodology; Jose M Polo, Resources, Supervision, Funding acquisition, Project administration; Minna-Liisa Anko, Conceptualization, Resources, Data curation, Formal analysis, Supervision, Funding acquisition, Validation, Investigation, Visualization, Methodology, Writing—original draft, Project administration, Writing—review and editing

## Author ORCIDs

Madara Ratnadiwakara (ID) http://orcid.org/0000-0001-7252-1823
Igor Ruiz De Los Mozos (ID) https://orcid.org/0000-0003-4097-6422
Minna-Liisa Anko (ID) http://orcid.org/0000-0003-0446-3566

## Ethics

Animal experimentation: All animal work was performed in strict accordance with the Australian code for the care and use of animals for scientific purposes (NHMRC) and the protocols were approved by the Monash University Animal Ethics Committee (MARP-2014-004).

## Decision letter and Author response

Decision letter https://doi.org/10.7554/eLife.37419.040
Author response https://doi.org/10.7554/eLife.37419.041

## Additional files

### Supplementary files

• Transparent reporting form
DOI: https://doi.org/10.7554/eLife.37419.028

### Data availability

Sequencing data sets have been deposited in GEO under accession codes GSE101905 and GSE113794. The iCLIP data has been made available in the public version of iCount (http://icount.biolab.si; search for SRSF3) and as source data to Figure 3.

The following datasets were generated:

| Author(s) | Year | Dataset title | Dataset URL | Database, license, and accessibility information |
|---|---|---|---|---|
| Anko M-L | 2018 | RNA sequencing of SRSF3 depleted pluripotent cells | https://www.ncbi.nlm.nih.gov/geo/query/acc.cgi?acc=GSE113794 | Publicly available at the NCBI Gene Expression Omnibus (accession no. GSE113794) |
| Buckberry S, Polo J, | 2017 | Transient and permanent | https://www.ncbi.nlm. | Publicly available at |

| Lister R, Knaupp A | | reconfiguration of chromatin and transcription factor occupancy drive reprogramming | nih.gov/geo/query/acc. cgi?acc=GSE101905 | the NCBI Gene Expression Omnibus (accession no. GSE10 1905) |
|---|---|---|---|---|
| Anko M-L | 2018 | iCLIP data from SRSF3 promotes pluripotency through Nanog mRNA export and coordination of the pluripotency gene expression program | http://icount.biolab.si | Available at iCount (SRSF3) |

The following previously published datasets were used:

| Author(s) | Year | Dataset title | Dataset URL | Database, license, and accessibility information |
|---|---|---|---|---|
| Hurt J, Robertson AD, Burge CB | 2013 | Global analysis of Upf1 in mESCs reveals expanded scope of nonsense-mediated mRNA decay | https://www.ncbi.nlm. nih.gov/geo/query/acc. cgi?acc=GSE41785 | Publicly available at the NCBI Gene Expression Omnibus (accession no. GSE41785) |
| Boutz PL, Sharp PA | 2015 | Detained introns are novel, widespread class of posttranscriptionally-spliced introns | https://www.ncbi.nlm. nih.gov/geo/query/acc. cgi?acc=GSE57231 | Publicly available at the NCBI Gene Expression Omnibus (accession no. GSE57231) |

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
