## [Decision Letter]

Thank you for submitting your work entitled "SRSF3 promotes pluripotency through Nanog mRNA export and coordination of the pluripotency gene expression program" for consideration by *eLife*. Your article has been reviewed by three peer reviewers, one of whom, Juan Valcárcel is a member of our Board of Reviewing Editors and the evaluation has been overseen by a Reviewing Editor and a Senior Editor.

The editors and referees agreed that there was too much further work needed to ask for a revised version at this point (see joint report below) and we are therefore returning your manuscript to you in case you wish to submit elsewhere. However, if you are willing and able to carry out the additional experiments and analysis requested, we would welcome the submission of a revised paper at a later date. This would be treated as a new submission with no guarantees of acceptance.

Summary:

Ratnadiwakara et al. further explore the role of the SR protein SRSF3 in reprogramming using a tamoxifen inducible Srsf3 mouse KO model in conjunction with a system for Dox induced expression of OKSM factors to reprogram MEFs to iPSCs. The study confirms the prior report by Ohta et al. showing that Srsf3 is induced during reprogramming and that depletion or ablation of Srsf3 reduces reprogramming efficiency as well as a loss of established pluripotency markers in iPSCs and loss of iPSC colony formation. The investigators note that there is a substantial reduction in protein levels of the pluripotency gene Nanog, but not mRNA in iPSCs following inducible Srsf3 KO. Follow up experiments supported the concept that Srsf3 promotes nuclear export of Nanog mRNA. Subsequent iCLIP experiments identify Nanog as a direct target of Srsf3 with binding sites in the first and last coding exon. A model is proposed where Srsf3 ablation leads to a loss of Nanog export. The iCLIP results also reveal other Srsf3 associated transcripts encoding pluripotency markers as well of regulators of chromatin state and other RNA binding proteins. Further studies explore the role of Srsf3 in regulating alternative splicing and stability of several targets, including the export receptor Nxf1, which had previously been shown to associate with Srsf3 and play a role in export. How this might be mechanistically linked to the effects on Nanog localization is not clear. Subsequent studies use RNA-Seq to identify changes in gene expression across the reprogramming time course and identify enrichment of Srsf3 binding features in or near several regulated exons and introns. mRNA-Seq was not carried out after Srsf3 depletion – which would have allowed more direct identification of Srsf3-regulated alternative splicing events. Since Srsf3 levels increase during reprogramming, it is reasonable to assume that a fraction of regulated events will be influenced by Srsf3. RNA features regulated during reprogramming and with associated Srsf3 CLIP tags were therefore identified ("SARFs"). A large fraction of SARFs showed intron binding – rather than the exon binding characteristic of splicing enhancement by Srsf3 – and many of the intron binding events overlapped with "detained introns" identified by Boutz and colleagues. The remaining experimental data looked into some examples of potential Srsf3 regulated splicing events with possible connections to the observed phenotype.

The manuscript is clearly written and presented, and the initial experiments setting up the iPS system with Srsf3 KO carefully conducted. The results are of interest because they emphasize the role of coordinated, multilayer post-transcriptional mechanisms of regulation in cellular pluripotency. The report confirms the important role of Srsf3 in pluripotency and development and has the potential to further inform our understanding of the mechanisms by identifying a number of different processes and targets that may be directly regulated by Srsf3. However, given that previous publications (adequately referenced by the authors) have already reported a role for SRSF3 in cell reprogramming (including effects on alternative splicing and nucleo-cytoplasmic transport of genes relevant for pluripotency, including Nanog), for publication in *eLife* the authors should make additional efforts to a) better document many of the experimental results, and b) consolidate the more novel aspects of their mechanistic findings (e.g. the role of SRSF3 on Nanog mRNA export).

Essential revisions:

1) Function of SRSF3 in mRNA export (Figure 4): further work is required to rigorously prove that SRSF3 binding is essential for Nanog mRNA export, that this is the main post-transcriptional mechanism of regulation of this transcript and that it has an impact on cell reprogramming. Nucleo-cytoplasmic distribution analysis of Nanog transcripts (Figure 4C) is not convincing, including the surprising equal partition of *Hprt* mRNAs, the apparent variability in the fraction of Nanog nuclear transcripts under normal conditions (0.4 in Figure 4C, 1.0 in Figure 4F), the relatively small (2-3 fold) and equal amount of nuclear and cytoplasmic mRNA in KO cells (which hardly seems to qualify as an export block) and, most importantly, the absence of standard marker analysis of the nucleus/cytoplasmic fractionation.

Furthermore, the experiments in Figure 4D and 4E are difficult to interpret as they are not explained in the text, legends, or Materials and methods section. Presumably these are overexpressed cDNAs? Were there any introns in the plasmid? If not, there might be concerns that the Srsf3-dependent export might be related to this artificial setting? Quantification of the total RNA levels of WT and ΔSRSF3 mRNAs should be shown, as well as statistical significance of differences in Figure 4E.

Given that Muller-McNicoll et al. already reported a role for SRSF3 in *Nanog* mRNA N/C transport, further demonstration that this is a limiting step in reprogramming would be expected. For example, is there evidence that the reduction in NANOG levels caused by the defect in transport is sufficient to compromise reprogramming (e.g. are ΔSRSF3 *Nanog* transcripts, expressed at levels comparable to endogenous *Nanog*, insufficient to mediate reprogramming)?

2) Documenting splicing changes associated with SRSF3 (Figure 5). The methods for quantitative analysis of splicing from RNA-Seq data and the subsequent follow up by qRT-PCR data in most cases did not allow assessment of the true degree of splicing change in KO cells or of the likelihood that identified events were truly connected to the phenotype. Specifically, none of the changes in splicing were expressed as changes in "percent spliced in" (PSI) or "percent intron retention" (PIR). Instead the focus was on statistically significant relative fold-changes of individual spliced junctions. Ideally, all the examples of regulated events should include quantitative information about the PSI changes observed.

Indeed from the genome browser views shown (Figure 5E,G and Figure 5—figure supplement 1E-H) only the increased Mbnl2 e6 inclusion (Figure 5E) was readily apparent by manual inspection. Even for this case, it would be helpful if the PSI values from RNA-Seq and from an RT-PCR spanning both introns flanking alt exon 6 (i.e. to look directly at inclusion versus skipping) were shown because the level of exon 6 in MEFs appears negligible, and hence it is not clear if a 3-fold increase in exon 6 splicing is quantitatively meaningful. It is also noted that the level of exon 6 inclusion in control cells does not change during the period of Srsf3 upregulation, calling into question the model where it promotes exon skipping. Figure 5E appears to show a substantial increase in PSI of Mbnl2 exon 6 between MEFs and iPSCs (judged from the relative height of the exon peaks – junction reads are not shown, Figure 5E and Figure 5—figure supplement 1F), as well as an increase in retention of introns 5 and 7. Yet in the time course in Panel F, the "exon inclusion" is quantitated by normalizing the exon 6-7 amplicon against *Hprt* and shows no change between day 1 and 16 (both are ~1.0). Perhaps this is because the qRT-PCR primers detect changes in both exon 6 PSI and expression level changes of Mbnl2, so that increased PSI is balanced by decreased overall Mbnl2 expression? If qRT-PCR is used to monitor relative changes in particular isoforms/alternative junctions, there should always be a separate primer pair across a constitutive junction in the same gene e.g. exon 7-8 in this case. A reciprocal primer pair should also ideally be used to assess the alternative "feature" – in this case Mbnl2 exon 6 skipping. Alternatively, RT-PCR using a single pair of primers in exons 5 and 7 could be used to give a direct read-out of PSI.

Similarly, the intron retention event in Nxf1 appears relatively low without an apparent difference in MEFs vs. iPSCs. This event does not appear in EST databases, raising the question as to whether the intron reads seen in the RNA-Seq represent unspliced introns that are nonetheless removed prior to export. While there is certainly a substantial reduction in intron 10 retention in response to Srsf3 KO (Figure 5H), intron retention appears to be a relatively minor pathway that does not appear to be strongly regulated during reprogramming (Figure 5—figure supplement E). Thus, the possible effect of Srsf3 to produce a possible truncated protein remains speculative and there is not sufficient evidence for "SRSF3 binding led to increased intron inclusion and production of a transcript variant encoding a short isoform of NXF1."

3) RNA-seq analyses. Unfortunately, the authors did not carry out a corresponding RNA-Seq analysis of gene expression and splicing in Srsf3 KO cells compared to controls at a relevant time point (e.g. day 16) to more directly validate the functionality of the Srsf3 binding targets by iCLIP. This is a necessary experiment and would be particularly valuable given that the authors did not have immunoprecipitation or input controls for the iCLIP which has been shown in several studies to be important to identify bona fide direct targets.

It would also be useful to summarize the types of RNA feature that are detected by JunctionSeq (it does not appear to consider the "classical" alternative splicing event types involving reciprocally changing features (cassette exon, retained intron etc), but rather individual splice junctions). It might be worth considering using one of the analysis pipelines that analyses "events" and provides a deltaPSI output (e.g. MISO, rMATS, MAJIQ…).

4) Data supporting a role for SRSF3 in regulating mRNA stability or other events (Figure 6). These data are either counterintuitive or are difficult to understand at the gene level. There are no experiments using standard mRNA stability assays (such as changes in half life after actinomycin D) and it is not clear in each of the cases shown how NMD would explain the differences in mRNA levels. The authors would need to show clearly how premature termination codons (PTCs) are being introduced in order to understand how NMD is involved. For *Ezh2* and *Dnmt3a* do they propose that an alteration in splicing is rendering the transcripts subject to NMD? If NMD is being invoked to explain the reduction in total mRNA in 6A, then it is not clear why both Ex 2-3 and Ex 19-20 primers would not show reduced levels in KO. In the case of *Epop*, there is no indication as to how the authors think NMD would account for the reduction in mRNA levels from this intronless gene with no PTC. Did the observed overlap of SRSF3 CLIP tags with detained intron events (Figure 5) and NMD-associated events (Figure 6) represent statistically significant enrichments?

---

## [Author Response]

Essential revisions:1) Function of SRSF3 in mRNA export (Figure 4): further work is required to rigorously prove that SRSF3 binding is essential for Nanog mRNA export, that this is the main post-transcriptional mechanism of regulation of this transcript and that it has an impact on cell reprogramming. Nucleo-cytoplasmic distribution analysis of Nanog transcripts (Figure 4C) is not convincing, including the surprising equal partition of Hprt mRNAs, the apparent variability in the fraction of Nanog nuclear transcripts under normal conditions (0.4 in Figure 4C, 1.0 in Figure 4F), the relatively small (2-3 fold) and equal amount of nuclear and cytoplasmic mRNA in KO cells (which hardly seems to qualify as an export block) and, most importantly, the absence of standard marker analysis of the nucleus/cytoplasmic fractionation.Furthermore, the experiments in Figure 4D and 4E are difficult to interpret as they are not explained in the text, legends, or Materials and methods section. Presumably these are overexpressed cDNAs? Were there any introns in the plasmid? If not, there might be concerns that the Srsf3-dependent export might be related to this artificial setting? Quantification of the total RNA levels of WT and ΔSRSF3 mRNAs should be shown, as well as statistical significance of differences in Figure 4E.Given that Muller-McNicoll et al. already reported a role for SRSF3 in Nanog mRNA N/C transport, further demonstration that this is a limiting step in reprogramming would be expected. For example, is there evidence that the reduction in NANOG levels caused by the defect in transport is sufficient to compromise reprogramming (e.g. are ΔSRSF3 Nanog transcripts, expressed at levels comparable to endogenous Nanog, insufficient to mediate reprogramming)?

First, we would like to clarify the experimental details of the *Nanog* mRNA export experiments that were carefully designed to control for any artefacts. These clarifications are also provided in the amended Results section and/or Materials and methods section of the manuscript. In experiments presented in Figure 4D-G, a WT-*Nanog* overexpression construct was used as a control to assure that the lack of introns or the level of expression did not have an impact on the interpretation of the results. Furthermore, primers specific for the overexpressed constructs were used and the same primers were used to quantify both WT-*Nanog* and ΔSRSF3-*Nanog* levels. As shown in the new supporting figure, the WT-*Nanog* and ΔSRSF3-*Nanog* mRNAs were expressed at equal levels that were comparable to the endogenous *Nanog* mRNA level (Figure 4—figure supplement 1). The WT-*Nanog* cDNA overexpressed in control pluripotent cells was exported similar to the endogenous *Nanog* mRNA (Figure 4C, F). In SRSF3-KO cells, endogenous *Nanog*, overexpressed WT-*Nanog* and ΔSRSF3-*Nanog* mRNAs were all retained in the nucleus. We also transfected the WT-*Nanog* and ΔSRSF3-*Nanog* constructs into HEK293 cells that do not express endogenous *Nanog* but do express SRSF3 and NXF1 and show that similar to pluripotent cells WT-*Nanog* mRNA is exported while the ΔSRSF3-*Nanog* is retained. We further show that NXF1 is not able to bind to *Nanog* mRNA lacking SRSF3 binding sites but binds to WT-*Nanog*. Importantly, the use of the cDNA constructs allowed us to show that the function of SRSF3 in mRNA export can be separated from its activity in splicing.

There are conflicting reports on the effect of NANOG on reprogramming outcome and its role during reprogramming, suggesting that the exact timing of NANOG overexpression is likely critical. As determining the time window for NANOG overexpression in reprogramming is beyond the scope of this manuscript, we did not perform the experiment overexpressing ΔSRSF3-*Nanog* during reprogramming. We would like to note that similar to the role of SRSF3 in mRNA export, the depletion of the THO export complex involved in the export of a different set of mRNAs in pluripotent cells and during reprogramming, is sufficient to induce differentiation and inhibit reprogramming. This suggests that mRNA export is broadly used as a post-transcriptional mechanism to regulate pluripotency. We have referenced the THO paper in the manuscript.

Specific questions regarding Figure 4C and F nucleo-cytoplasmic distribution analysis: The values in 4C have been normalised to Ctrl (=1), so they are relative values. Similarly, in 4F the values have been normalised to Ctrl. Thus, values in Figures 4C and 4F are not directly comparable. This also means that *Hprt* levels are not equal in the nucleus and cytoplasm. Of note, at least for *Actb* mRNA in Mueller-McNicoll et al., 2017, the mature mRNA levels are equal in the nucleus and cytoplasm (Figure 4—figure supplement 1B of the paper). We have included the standard marker analysis for nucleo-cytoplasmic fractionation in Figure 4—figure supplement 1. We have included more detail in the text and figure legends to clarify the experimental design and data analysis. We have added the total mRNA quantifications in Figure 4—figure supplement 1.

2) Documenting splicing changes associated with SRSF3 (Figure 5). The methods for quantitative analysis of splicing from RNA-Seq data and the subsequent follow up by qRT-PCR data in most cases did not allow assessment of the true degree of splicing change in KO cells or of the likelihood that identified events were truly connected to the phenotype. Specifically, none of the changes in splicing were expressed as changes in "percent spliced in" (PSI) or "percent intron retention" (PIR). Instead the focus was on statistically significant relative fold-changes of individual spliced junctions. Ideally, all the examples of regulated events should include quantitative information about the PSI changes observed.Indeed from the genome browser views shown (Figure 5E,G and Figure 5—figure supplement 1E-H) only the increased Mbnl2 e6 inclusion (Figure 5E) was readily apparent by manual inspection. Even for this case, it would be helpful if the PSI values from RNA-Seq and from an RT-PCR spanning both introns flanking alt exon 6 (i.e. to look directly at inclusion versus skipping) were shown because the level of exon 6 in MEFs appears negligible, and hence it is not clear if a 3-fold increase in exon 6 splicing is quantitatively meaningful. It is also noted that the level of exon 6 inclusion in control cells does not change during the period of Srsf3 upregulation, calling into question the model where it promotes exon skipping. Figure 5E appears to show a substantial increase in PSI of Mbnl2 exon 6 between MEFs and iPSCs (judged from the relative height of the exon peaks – junction reads are not shown, Figure 5E and Figure 5—figure supplement 1F), as well as an increase in retention of introns 5 and 7. Yet in the time course in Panel F, the "exon inclusion" is quantitated by normalizing the exon 6-7 amplicon against Hprt and shows no change between day 1 and 16 (both are ~1.0). Perhaps this is because the qRT-PCR primers detect changes in both exon 6 PSI and expression level changes of Mbnl2, so that increased PSI is balanced by decreased overall Mbnl2 expression? If qRT-PCR is used to monitor relative changes in particular isoforms/alternative junctions, there should always be a separate primer pair across a constitutive junction in the same gene e.g. exon 7-8 in this case. A reciprocal primer pair should also ideally be used to assess the alternative "feature" – in this case Mbnl2 exon 6 skipping. Alternatively, RT-PCR using a single pair of primers in exons 5 and 7 could be used to give a direct read-out of PSI.Similarly, the intron retention event in Nxf1 appears relatively low without an apparent difference in MEFs vs. iPSCs. This event does not appear in EST databases, raising the question as to whether the intron reads seen in the RNA-Seq represent unspliced introns that are nonetheless removed prior to export. While there is certainly a substantial reduction in intron 10 retention in response to Srsf3 KO (Figure 5H), intron retention appears to be a relatively minor pathway that does not appear to be strongly regulated during reprogramming (Figure 5—figure supplement 1E). Thus, the possible effect of Srsf3 to produce a possible truncated protein remains speculative and there is not sufficient evidence for "SRSF3 binding led to increased intron inclusion and production of a transcript variant encoding a short isoform of NXF1."

To quantify splicing changes following SRSF3 depletion, we have used MISO and expressed the splicing changes as delta-PSI or PIR. We have used PSI or PIR values in all figures when possible. For the qPCR analysis of *Nxf1* intron retention, we expressed the data as% -retained (retained/total) which takes into account any changes in expression. We have used both MISO and JunctionSeq analysis in the context of the reprogramming time course (Figure 5 and Figure 6), as well as compared the two. To further consolidate the data, we have now focused on *Nxf1* splicing regulation and provide other regulated events as further examples of the dynamic SRSF3 mediated splicing during reprogramming. We have included in the Figure 6—figure supplement 1 data on previously identified transcripts with intron detention and show that we can detect these events, and in fact, *Clk1* and *Mdm4* that were previously investigated by Boutz et al. are regulated during reprogramming. This provides further support for the dynamic changes in intron detention during reprogramming, and the involvement of SRSF3 in the splicing of detained introns and exons flanking them.

Specific queries regarding Figure 5 (current Figures 5 and 6): Although PIR or PSI differences between MEFs and iPSC were not large, there was in each case a significant change in the course of reprogramming (for example, *Nxf1* intron retention increased from PIR 0.1 to 0.4 by day 12 which exceeds the commonly used significance criterion of 0.2). Furthermore, by only comparing MEFs to iPSC many important regulatory events are clearly missed and RNA processing changes during cell transitions need to be characterised throughout the process. There are a number of published papers (that we cite) on the *Nxf1* intron retention event and it is annotated in multiple genome databases so it is surprising if it does not appear in EST databases. It is hard to define what level of splicing is biologically meaningful and any cut-off is arbitrary. We feel that the strong loss-of-pluripotency phenotype is the best evidence for the biological significance of SRSF3 mediated splicing regulation during reprogramming and in pluripotent cells.

3) RNA-seq analyses. Unfortunately, the authors did not carry out a corresponding RNA-Seq analysis of gene expression and splicing in Srsf3 KO cells compared to controls at a relevant time point (e.g. day 16) to more directly validate the functionality of the Srsf3 binding targets by iCLIP. This is a necessary experiment and would be particularly valuable given that the authors did not have immunoprecipitation or input controls for the iCLIP which has been shown in several studies to be important to identify bona fide direct targets.It would also be useful to summarize the types of RNA feature that are detected by JunctionSeq (it does not appear to consider the "classical" alternative splicing event types involving reciprocally changing features (cassette exon, retained intron etc.), but rather individual splice junctions). It might be worth considering using one of the analysis pipelines that analyses "events" and provides a deltaPSI output (e.g. MISO, rMATS, MAJIQ…).

We have now included the RNA-sequencing of SRSF3 deficient cells in the manuscript. We analysed the data using MISO and show that substantial splicing changes take place following SRSF3 depletion. We show the different types of events that are affected, provide examples of regulated events as well as a global map of SRSF3 binding sites over skipped exons affected by SRSF3 ablation. We focused our analysis on skipped exons because this is the largest group annotated and assessed by MISO and thus provided a sufficient number of data points for reliable analysis. The ‘SRSF3 RNA-map’ is first of its kind as previous SRSF3 CLIP-seq or RNA-seq studies have not mapped the binding sites to regulated exons.

We have now analysed the reprogramming time course RNA-seq data using MISO and JunctionSeq. We demonstrate that AS is affected across event types during reprogramming. We have included the proportions of the different types of AS events during the reprogramming time course. We originally chose to use JunctionSeq because it allowed an unbiased assessment of both annotated and novel RNA processing changes. Our focus was to determine the localisation of SRSF3 binding sites relative to splice junctions that change during reprogramming and therefore JunctionSeq was a good alternative. This analysis captured the previously reported cumulative increase in alternative splice site usage from MEFs to iPSCs (Ohta et al., 2013 and Cieply et al., 2016) and readily captured RNA processing changes across expression levels and independent of gene expression changes (Figure 6—figure supplement 1). The re-analysis using MISO did not change our interpretations. Importantly, it further confirmed that most splicing changes take place at day 3 and iPSC, and that the comparison of iPSCs to MEFs represents a cumulative increase in AS, the transition from day 12 to iPSC contributing most events.

4) Data supporting a role for SRSF3 in regulating mRNA stability or other events (Figure 6). These data are either counterintuitive or are difficult to understand at the gene level. There are no experiments using standard mRNA stability assays (such as changes in half life after actinomycin D) and it is not clear in each of the cases shown how NMD would explain the differences in mRNA levels. The authors would need to show clearly how premature termination codons (PTCs) are being introduced in order to understand how NMD is involved. For Ezh2 and Dnmt3a do they propose that an alteration in splicing is rendering the transcripts subject to NMD? If NMD is being invoked to explain the reduction in total mRNA in 6A, then it is not clear why both Ex 2-3 and Ex 19-20 primers would not show reduced levels in KO. In the case of Epop, there is no indication as to how the authors think NMD would account for the reduction in mRNA levels from this intronless gene with no PTC. Did the observed overlap of SRSF3 CLIP tags with detained intron events (Figure 5) and NMD-associated events (Figure 6) represent statistically significant enrichments?

We have now clarified this section. We have included the analysis of differential gene expression in SRSF3 depleted cells as well as compared DEGs to the data from Hurt et al., 2013. We have re-worded our conclusions based on the current Figure 7 to state that SRSF3 is involved in controlling the steady-state levels of distinct mRNAs regulated by NMD in pluripotent cells. Our data and the data by Hurt et al. show that in many cases only one mRNA (called isoform in Hurt et al.) is affected by NMD inhibition, suggesting that transcripts produced from a single gene can be differentially regulated. It is important to note that these mRNAs (isoforms) do not necessarily contain PTCs. Multiple recent studies (for instance, Imamachi et al., 2017, Colombo et al., 2017, Hurt et al., 2013 and Zund et al.) demonstrate that besides degrading aberrant mRNAs that harbor PTCs, NMD also targets many seemingly "normal" mRNAs that encode full-length proteins. Further studies are required to fully define how the NMD machinery recognises these mRNAs. We show that SRSF3 specifically binds to and regulates the abundance of NMD-sensitive mRNAs (isoforms).

Specific queries regarding Figure 7: We have now determined the half-lives of the mRNAs as in Ke et al., 2016 and added n values or statistics where relevant.